# Transposon-activated *POU5F1B* promotes colorectal cancer growth and metastasis

Laia Simó-Riudalbas [1] ✉, Sandra Offner[1], Evarist Planet[1], Julien Duc [1], Laurence Abrami [1], Sagane Dind[1], Alexandre Coudray[1], Mairene Coto-Llerena [2,3], Caner Ercan [2,3], Salvatore Piscuoglio [2,3], Claus Lindbjerg Andersen [4], Jesper Bertram Bramsen [4] & Didier Trono [1] ✉

The treatment of colorectal cancer (CRC) is an unmet medical need in absence of early diagnosis. Here, upon characterizing cancer-specific transposable element-driven transpochimeric gene transcripts (TcGTs) produced by this tumor in the SYSCOL cohort, we find that expression of the hominid-restricted retrogene *POU5F1B* through aberrant activation of a primate-specific endogenous retroviral promoter is a strong negative prognostic biomarker. Correlating this observation, we demonstrate that POU5F1B fosters the proliferation and metastatic potential of CRC cells. We further determine that POU5F1B, in spite of its phylogenetic relationship with the POU5F1/OCT4 transcription factor, is a membrane-enriched protein that associates with protein kinases and known targets or interactors as well as with cytoskeleton-related molecules, and induces intracellular signaling events and the release of *trans*-acting factors involved in cell growth and cell adhesion. As *POU5F1B* is an apparently non-essential gene only lowly expressed in normal tissues, and as *POU5F1B*-containing TcGTs are detected in other tumors besides CRC, our data provide interesting leads for the development of cancer therapies.

The human genome contains more than 4.5 million inserts derived from transposable elements (TEs). This so-called endovirome is a major motor of genome evolution due its gene-disruptive and recombinogenic potential and because it harbors a high density of transcription factors binding sites[1,2]. TE-embedded regulatory sequences (TEeRS) can act as promoters, enhancers, repressors, terminators or insulators and, as such, exert profound influences on human development and physiology, from embryonic genome activation to metabolic control, and from brain development to innate immunity[3,4]. Furthermore, the TE-based regulome confers a high degree of species specificity to the conduct of biological events due to the rapid evolutionary turnover of both its TE constituents and their sequence-specific *trans*-acting controllers[4,5].

Alterations of this regulatory system can lead to cancer, as was immediately suggested by the phylogenetic relationship between endogenous retroviruses (ERV) and their exogenous relatives, the long-called RNA tumor viruses that led to the discovery of oncogenes[6]. Since then, TEs have been linked to human cancer through a variety of observations and mechanisms[7], including the promotion of chromosomal rearrangements owing to their repetitiveness[8], the inactivation of tumor suppressor genes through insertional mutagenesis[9], and the production of long non-coding RNAs or chimeric gene transcripts from aberrantly activated TEeRS[10–12]. A wide range of oncogene-encoding TE-driven transpochimeric gene transcripts (TcGTs) have been documented in a recent survey of cancer databases[13]. However, a causal role for these products in oncogenesis has so far seldom been demonstrated.

Colorectal cancer is the second most frequently encountered non-skin malignancy in both women and men. This tumor is increasingly diagnosed in young people, and its treatment is a largely unmet

[1]School of Life Sciences, Ecole Polytechnique Fédérale de Lausanne, Lausanne, Switzerland. [2]Institute of Medical Genetics and Pathology, University Hospital Basel, 4031 Basel, Switzerland. [3]Visceral Surgery Research Laboratory, Department of Biomedicine, University of Basel, Basel, Switzerland. [4]Department of Molecular Medicine, Aarhus University Hospital, DK8200 Aarhus N, Denmark. ✉e-mail: laia.simoriudalbas@epfl.ch; didier.trono@epfl.ch

clinical need unless it is detected at an early stage (https://www.cancer.org/cancer/colon-rectal-cancer/about/key-statistics.html). Facilitating molecular explorations, its step-wise development has been well documented, and its surgical resection commonly provides an abundance of clearly distinct normal, tumoral and sometimes metastatic tissue for in-depth molecular analyzes. Here, we used analytical pipelines tailored for the transposcriptome, or sum of TE-derived RNA transcripts present in a cell, to RNA sequencing data obtained from a large cohort of CRC patients. This led us to identify the aberrant TE-driven expression of *POU5F1B*, a hominid-restricted retrogene, as a strong negative predictor of the clinical course of this malignancy and to determine that its POU5F1B product is responsible for fostering the growth and metastatic potential of colorectal cancer cells through a combination of *cis*- and *trans*-effects.

## Results

### Cancer-specific TcGTs in CRC

We sought potentially oncogenic TcGTs through the in-depth analysis of previously published RNA-seq data from primary tumors and matched normal colon of 301 patients diagnosed with colorectal adenoma (11%) or adenocarcinoma (89%), referred to hereafter as the SYSCOL cohort[14]. High-quality sequencing and histopathological data were available for 286 of the 301 patients. Ninety-five TcGTs involving 39 genes were detected in more than 20% of tumors and less than 10% of non-tumoral tissue samples (Table 1). About half (44/95) of these TcGTs were predicted to encode the canonical protein and the rest to yield truncated (43) or out-of-frame (8) derivatives (Table 1 and Supplementary Data 1). TcGTs involving only three of these 39 genes (*BMP7*, *RNF43,* and *SLCO1B3*) had previously been identified in cancer[13,15].

The tumor-restricted TcGT most frequently detected in our cohort initiated within a primate-specific *LTR66* endogenous retroviral promoter on chromosome 8q24, spliced into several other TE inserts (*AluSx1*, *LTR33*, *L2b*, and *MLT1F1*) and ended in *POU5F1B*, a paralog of the *POU5F1/OCT4* pluripotency gene. *POU5F1B*-containing TcGTs, about two-thirds starting at *LTR66*, were found in 186 (65%) CRC primary tumors *versus* 11 (3.8%) samples labeled as normal colon (Fig. 1a 'cohort 1', Table 1, Supplementary Fig. 1a, b). Other forms of *POU5F1B* TcGTs starting from some of the intervening TEs were also noted in some tumors, but all were distinct from the annotated *POU5F1B* mono-exonic transcript driven by a promoter partly overlapping with the *MLT1F1* sequence (Supplementary Fig. 1b and Supplementary Data 1). We validated these results with another publicly available dataset of 18 CRC patients (GSE50760)[16], where the *LTR66-POU5F1B* TcGT was found in 0/18 normal colons, 10/18 primary tumors, and 12/18 liver metastases (Fig. 1a 'cohort 2'). A re-analysis of single-cell RNA-seq (scRNA-seq) data from primary tumors and nearby normal mucosa in 11 CRC patients[17] further revealed that *LTR66* and *POU5F1B* were expressed in epithelial cells and only exceptionally detected in other cell types (Supplementary Fig. 1c). Confirming these data, a RACE (5'-rapid amplification of cDNA ends) analysis documented the presence of a non-canonical ~2,500 bp *POU5F1B* TcGT starting within *LTR66* in several CRC cell lines (Supplementary Fig. 1d). In CRC RNA-seq datasets from the SYSCOL cohort (Fig. 1b), the CRC TCGA collection and 54 CRC cell lines from EGAD00001000725[18] (Supplementary Fig. 1e), high levels of *LTR66*- and *POU5F1B*-derived RNAs systematically correlated, indicating that they likely corresponded in their majority to *LTR66-POU5F1B* TcGTs.

### *LTR66-POU5F1B* is a negative CRC prognostic marker in the SYSCOL cohort

We noted that increased expression of *LTR66-POU5F1B* in the SYSCOL cohort coincided with the transition from normal tissue to adenoma and from adenoma to carcinoma (Fig. 1c), which strongly suggested

**Table 1 | Thirty-nine TcGT-transcribed genes were detected in more than 20% of CRC tumors and less than 10% of normal colons**

| Gene | nN | nT | %N | %T | Protein product |
|---|---|---|---|---|---|
| POU5F1B | 11 | 186 | 3.8 | 65.0 | canon. |
| HTR1D | 9 | 175 | 3.1 | 61.2 | canon. |
| SLCO1B3 | 6 | 159 | 2.1 | 55.6 | N-trunc. |
| CMTR1 | 2 | 149 | 0.7 | 52.1 | C-trunc. |
| ABLIM2 | 1 | 111 | 0.3 | 38.8 | N-ext. |
| NCALD | 28 | 124 | 9.8 | 43.4 | canon. |
| SLC22A3 | 25 | 119 | 8.7 | 41.6 | N-trunc.; N-C-trunc. |
| PMPCB | 7 | 94 | 2.4 | 32.9 | N-C-trunc.; out.-frame; C-trunc. |
| CAPN12 | 2 | 75 | 0.7 | 26.2 | canon.; C-trunc. |
| FGGY | 8 | 80 | 2.8 | 28.0 | canon.; N-trunc. |
| TRAF2 | 8 | 80 | 2.8 | 28.0 | canon.; N-ext. |
| ATP5J2 | 13 | 85 | 4.5 | 29.7 | canon. |
| RNF43 | 3 | 71 | 1.0 | 24.8 | canon.; N-ext. |
| EDAR | 0 | 66 | 0.0 | 23.1 | N-ext. |
| TM9SF2 | 4 | 70 | 1.4 | 24.5 | canon. |
| CEP72 | 25 | 91 | 8.7 | 31.8 | N-trunc.; N-C-trunc. |
| TSSC1 | 9 | 73 | 3.1 | 25.5 | N-trunc.; C-ext. |
| KRT8 | 7 | 67 | 2.4 | 23.4 | N-ext. |
| WFDC3 | 24 | 83 | 8.4 | 29.0 | N-trunc. |
| PALD1 | 16 | 74 | 5.6 | 25.9 | canon.;C-trunc.; out.frame |
| SBF2 | 12 | 69 | 4.2 | 24.1 | out.frame; N-trunc. |
| SPACA3 | 3 | 58 | 1.0 | 20.3 | N-ext. |
| MCCC2 | 6 | 61 | 2.1 | 21.3 | out.frame; N-C-trunc. |
| POMZP3 | 20 | 70 | 7.0 | 24.5 | canon.; C-trunc. |
| TNRC18 | 22 | 72 | 7.7 | 25.2 | N-trunc.; N-C-trunc. |
| BMP7 | 12 | 62 | 4.2 | 21.7 | out.frame |
| ABHD2 | 13 | 63 | 4.5 | 22.0 | canon. |
| TGIF2 | 14 | 63 | 4.9 | 22.0 | canon. |
| ST3GAL2 | 23 | 70 | 8.0 | 24.5 | canon. |
| ZNF283 | 19 | 66 | 6.6 | 23.1 | canon.; out.frame |
| ADAP1 | 16 | 63 | 5.6 | 22.0 | canon. |
| CLSTN3 | 23 | 66 | 8.0 | 23.1 | canon. |
| CLDN4 | 16 | 58 | 5.6 | 20.3 | canon. |
| FAM110A | 17 | 59 | 5.9 | 20.6 | canon. |
| DAP3 | 27 | 67 | 9.4 | 23.4 | N-trunc.; out.frame |
| PTPDC1 | 27 | 67 | 9.4 | 23.4 | canon.; N-trunc. |
| PMFBP1 | 20 | 58 | 7.0 | 20.3 | canon.; C-trunc.; N-C-trunc. |
| TXLNG | 26 | 63 | 9.1 | 22.0 | N-C-trunc. |
| ZNF710 | 26 | 60 | 9.1 | 21.0 | N-ext. |

*nN* number of normal samples, *nT* number of tumors, *%N* percentage in normal colon, *%T* percentage in tumors, *canon.* canonical protein, *N-trunc.* N-terminal truncated protein, *C-trunc.* C-terminal truncated protein, *N-C-trunc.* N-C-terminal truncated protein, *N-ext.* N-terminal extended protein, *out.frame* out of frame protein.

that *LTR66-POU5F1B* was a marker of advanced disease. Confirming this hypothesis, a Kaplan-Meier representation of overall survival and a Cox Proportional Hazards survival analysis of the SYSCOL cohort revealed that this TcGT was associated with lower overall survival ($P = 0.0007$; hazard ratio (HR) = 2.52, 95% CI = 1.41–4.51) (Fig. 1d) and shorter relapse-free survival ($P = 0.038$; HR = 8.61, 95% CI = 1.15–64.71) (Fig. 1e) in stage II and III patients. The negative prognostic value of POU5F1B was also significant when stage I and IV patients were

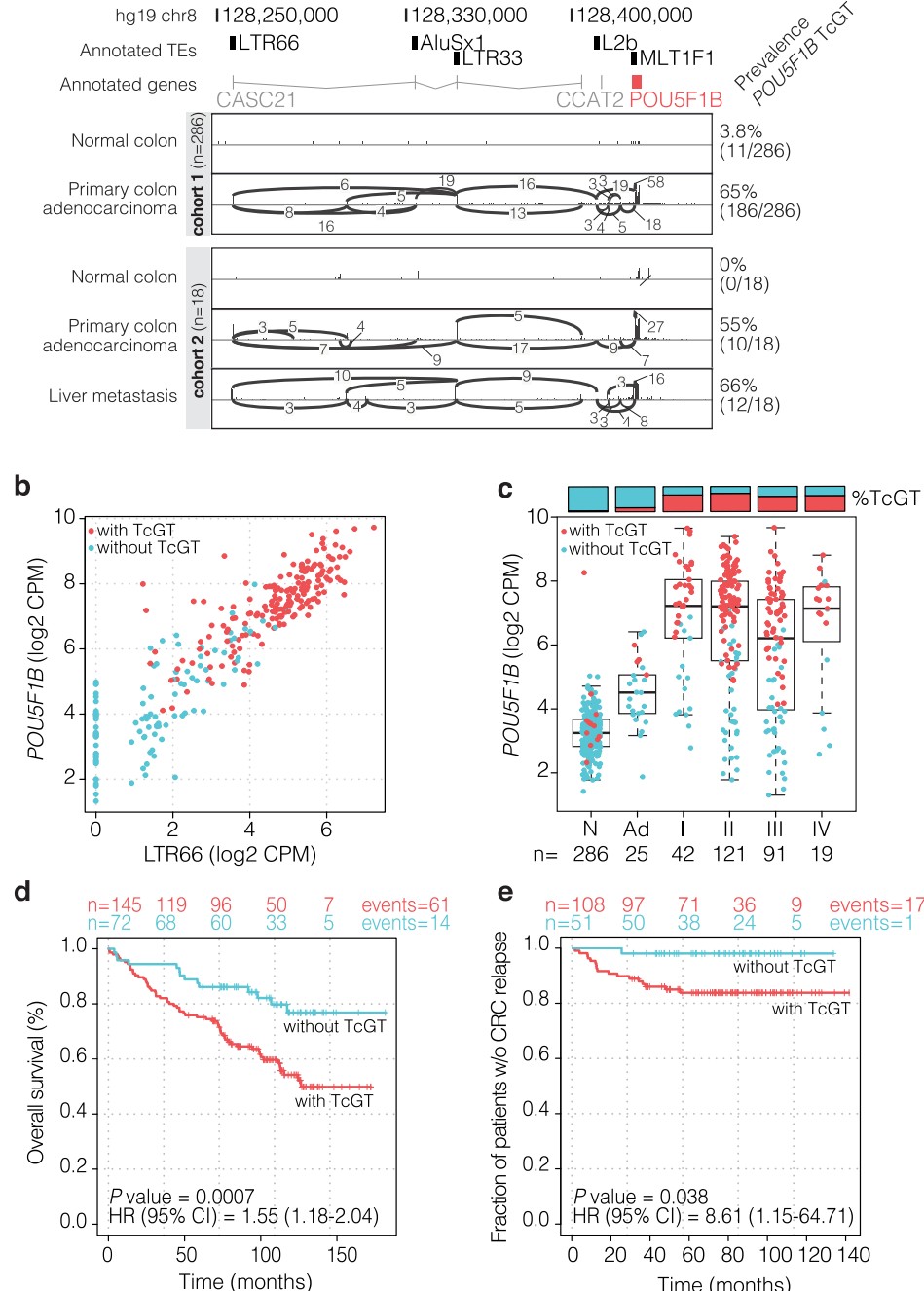

**Fig. 1 | An LTR66 endogenous retroviral promoter drives POU5F1B expression in CRC. a** Chromosome 8q24.21 genomic locus, with transposable elements (TEs) (black boxes), non-coding RNAs CASC21 and CCAT2 (gray lines) and POU5F1B gene (red box). Splice junctions were highlighted in two representative CRC patients, respectively, from the SYSCOL (cohort 1) and GSE50760 (cohort 2) cohorts. **b** Correlation of LTR66 and POU5F1B RNA levels in SYSCOL CRC RNA-seq dataset (*n* = 286 patients, two-sided Pearson cor = 0.91, *P* = 1e-16). **c** POU5F1B expression levels in normal colon (N), adenoma (Ad), and stage I-IV carcinoma samples from the SYSCOL cohort, as measured by RNA-seq. Boxplots are represented as first and third quartiles with a median in the center. Whiskers are defined as 1.5 times the interquartile range. **d** Kaplan–Meier representation of overall survival in stages II-III SYSCOL patients according to the presence (*n* = 145) or absence (*n* = 72) of POU5F1B TcGT. **e** Kaplan-Meier representation of relapse-free survival in stage II-III SYSCOL patients with respect to the presence (*n* = 108) or absence (*n* = 51) of POU5F1B TcGTs. In **d**, **e** HR and CI were computed by fitting the univariate Cox model. Source data are provided as a Source Data file.

included (*P* = 0.004; HR = 1.32, 95% CI = 1.08–1.61) but their numbers (*n* = 42 and *n* = 19, respectively) were insufficient to conclude that POU5F1B was also influential at these stages. Importantly, POU5F1B-encoding TcGTs were expressed in both microsatellite stable (MSS) and microsatellite unstable (MSI) tumors, and in all four CRC consensus molecular subtypes (CMS)[19] (Supplementary Fig. 1f), although they were found more frequently in the MSS and CMS2 subsets. CMS2

tumors are characterized by marked WNT and MYC signaling activation[19], and the *MYC* coding sequence is located some 300 kb downstream of *POU5F1B*. Yet, while there was a significant correlation between *POU5F1B* and *MYC* expression in CRC cell lines and tumor samples, the presence of many outliers indicated that the two genes were not systematically coregulated (Supplementary Fig. 1g). Furthermore, while TcGT-driven *POU5F1B* overexpression was associated

with shorter survival in stages II and III patients, this was not the case for *MYC* according to either univariate or *POU5F1B/MYC* multivariate analyzes (Supplementary Fig. 1h). Of note, whereas these data established the *LTR66-POU5F1B* TcGT as a negative CRC prognostic marker in the SYSCOL cohort, we did not observe this correlation to be significant in the TCGA dataset (Supplementary Fig. 1i).

## POU5F1B enhances the growth and metastatic potential of CRC cells

To investigate the functional consequences of POU5F1B upregulation in colorectal cancer, we used a combination of in vitro and in vivo studies. Stable transduction with a lentiviral vector expressing an HA-tagged form of POU5F1B (Fig. 2a) stimulated the colony forming and proliferation abilities of SW480 (Fig. 2b, c), HT29, and SW620 (Supplementary Fig. 2a, b) cells. Moving to xenotransplantation experiments, we found that subcutaneous injection of luciferase-containing POU5F1B-overexpressing SW480 cells in NOD/SCID/gamma (NSG) immunodeficient mice induced the formation of tumors that were bigger and heavier than obtained with GFP-overexpressing controls (Fig. 2d). When equal-size fragments of these tumors were then implanted in the cecum of another set of NSG mice, total body bioluminescence monitoring yielded higher signals in SW480-POU5F1B-versus SW480-GFP-engrafted animals (Fig. 2e). The transplanted tumor fragments themselves presented roughly similar masses and volumes at the implantation site (Supplementary Fig. 2c), suggesting that the fitness of well-constituted tumors may be less critically dependent on POU5F1B than their initial establishment and growth. However, SW480-POU5F1B recipient mice presented greater numbers of macroscopically visible metastases in the liver and of additional tumor-invaded organs (Fig. 2f), which explained their overall higher level of bioluminescence (Fig. 2e). Finally, when POU5F1B-overexpressing SW480 cells were injected in the spleen of NSG mice and these animals splenectomized rapidly thereafter, they induced markedly higher numbers of liver metastases than their GFP-expressing controls (Fig. 2g). To extend these data, we repeated the intra-splenic injection experiment with similarly modified SW620 CRC cells, and observed that POU5F1B-overexpressing cells yielded higher densities of liver metastases and greater numbers of extra-hepatic metastases than control cells (Supplementary Fig. 2d).

We then turned to in vivo loss-of-function experiments using *POU5F1B*-expressing CRC cell lines. Downregulating *POU5F1B* by dox-inducible lentivector-mediated RNA interference impaired the growth of LS1034 (Fig. 2h, Supplementary Fig. 2e), HT55 and LS174T CRC cells (Supplementary Fig. 2f, g), all of which express this TcGT at baseline. Mice kept on doxycycline for one week beforehand were then injected subcutaneously with dox-treated control or POU5F1B knockdown LS1034 cells. Tumor growth was markedly decreased after injection of POU5F1B-depleted cells, confirming the pro-oncogenic effect of this factor (Fig. 2i). Furthermore, following intrasplenic injection and splenectomy, LTR66-POU5F1B knockdown LS1034 cells also displayed a decreased metastatic phenotype both in and out of the liver, compared with control cells (Supplementary Fig. 2h).

## *LTR66-POU5F1B* TcGTs result from aberrant activation of an intronic enhancer

To determine what triggered the production of *LTR66-POU5F1B* TcGTs, we analyzed molecular changes occurring at their source locus, located in a gene desert that harbors previously identified risk loci for several epithelial cancers such as the GWAS prostate and colorectal cancer-associated SNP rs6983267[20–22]. This SNP resides 15 kb upstream of *POU5F1B* in a regulatory sequence previously described as an enhancer for *MYC*[23,24]. For samples for which sequencing depth allowed an assessment of the allelic mode of expression of *LTR66-POU5F1B*, it was found to be biallelic in a significant fraction (Supplementary Fig. 3a). In addition, detection of

this TcGT correlated neither with the rs6983267 genotype in the SYSCOL tumors nor with copy number variations (CNVs) in the TCGA CRC database (Supplementary Fig. 3b, c). We thus turned to chromatin analyzes. By exploring publicly available data, we found that the rs6983267-containing locus physically interacts with *LTR66* in three CRC cell lines[25] and that both regions are characterized by ATAC-seq (transposase-accessible chromatin sequencing) peaks in CRC tumor samples from TCGA[26] (Fig. 3a and Supplementary Fig. 3d). This suggested that the rs6983267-containing sequence acts as an *LTR66-POU5F1B* intronic enhancer. Accordingly, this locus was enriched in the activation marks H3K4me1 and H3K27ac in LS1034 and HT55, two CRC cell lines expressing the TcGT, but not in SW480, where *LTR66-POU5F1B* is not detected (Fig. 3c and Supplementary Fig. 1e). Correspondingly, the neighboring *LTR66* was adorned with the active promoter marks H3K4me3 and H3K27ac in the first two but not the third of these cell lines (Fig. 3b). In HT29 cells, where RNA-seq detected low levels of *LTR66* and *POU5F1B* reads (Supplementary Fig. 1e), some H3K4me3 was detected at *LTR66* whereas the intronic enhancer region was more enriched in H3K4me1 and H3K27ac (Fig. 3b, c). We then manipulated the chromatin status at *LTR66* and its putative enhancer through gRNA-targeting of a dCas9-KRAB fusion protein (CRISPRi)[27], which induces deposition of the repressive mark H3K9me3, or of its dCas9-VPR counterpart (CRISPRa)[28], which activates transcription (Fig. 3d). Expression of the *LTR66-POU5F1B* TcGT was decreased in LS1034 and HT55 cells when CRISPRi was targeted either to its *LTR66* promoter or to its putative intronic enhancer (Fig. 3e, f, upper plots). Inversely, it was induced by targeting CRISPRa to *LTR66* and, modestly, to the TcGT internal enhancer in HT29 cells (Fig. 3e, f, lower plots).

## POU5F1B is a hominoid-restricted OCT4 paralog localizing to cytoplasm and enriched in membranes

*POU5F1B* results from the retrotransposition of *POU5F1/OCT4*, the stem cell pluripotency gene, in the last common ancestor of modern Hominidae (Supplementary Fig. 4a). In human, the products of these two paralogs differ at 15 out of 359/360 amino acid positions (Supplementary Fig. 4b). However, immunofluorescence microscopy revealed that, contrasting with the exclusively nuclear OCT4, POU5F1B was detected only at low levels in this compartment and instead strongly accumulated in the cytoplasm of overexpressing SW480 cells (Fig. 4a and Supplementary Fig. 4c). Of note, the cytoplasmic enrichment of POU5F1B was not cell type–specific, as it was also apparent in transfected 293 T cells (Supplementary Fig. 4d). This preferential subcellular localization was confirmed by immunohistochemistry analysis of endogenous POU5F1B in primary CRC biopsy samples, where a positive signal strictly correlating with the detection of POU5F1B transcripts by RT-PCR was detected in 4 out of 5 examined tumors but none of their matching normal tissue counterparts (Fig. 4b and Supplementary Fig. 4e, g). Cytoplasmic endogenous POU5F1B was also detected in LS1034 and HT55 cells (Supplementary Fig. 4f). Furthermore, the subcellular fractionation of SW480 cells overexpressing HA-tagged POU5F1B demonstrated that it was cytoplasmic and enriched in membranes, whereas OCT4 displayed a preference for the nucleus (Fig. 4c). Accordingly, chromatin immunoprecipitation studies failed to document significant genomic recruitment for POU5F1B (not illustrated). Isolation of detergent-resistant membranes (DRMs) from POU5F1B- and GFP-overexpressing SW480 cells further revealed that POU5F1B was enriched in this fraction (F2) (Fig. 4d), which interestingly comprises cholesterol-sphingolipid rafts where signaling complexes are commonly assembled. Of note, a fraction of transferrin receptor was also found to relocate partly to DRMs in POU5F1B-overexpressing cells, suggesting that the TcGT product perhaps exerted some restructuring influence on membranes.

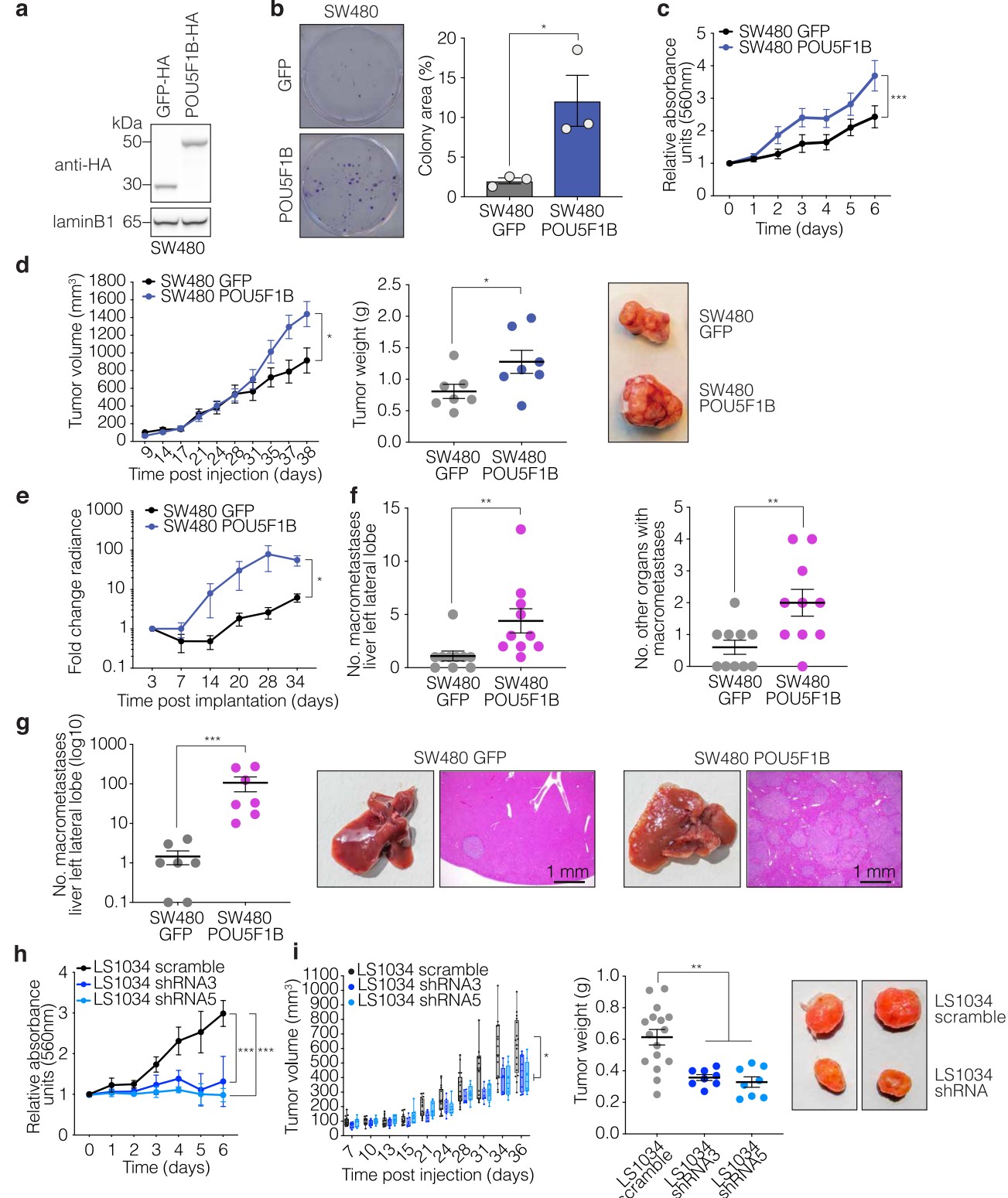

## Multi-omic characterization of POU5F1B-induced molecular changes

To define the impact of *POU5F1B* activation, we first analyzed RNA-seq data from *GFP*- and *POU5F1B*-overexpressing HT29 and SW480 cells, from control and *POU5F1B*-knockdown LS1034 cells, and from stages I, II, and III SYSCOL CRC primary tumors (Supplementary Fig. 5a, b). Of note, the transcriptional changes associated with POU5F1B expression were globally mild when compared with those triggered by POU5F1/OCT4, as illustrated by a side-by-side comparison of HT29 cells

modified to overexpress either one of these two paralogs (Supplementary Fig. 5a). For the 3 settings (SW480, LS1034, SYSCOL) with enough upregulated genes in the presence of *POU5F1B*, delineation of GO terms using the Functional Annotation Chart from DAVID bioinformatics resources 6.8[29] consistently pointed to the term 'extracellular space' (Supplementary Fig. 5b). Several genes stood out, albeit not systematically in all 4 settings, which encoded for secreted proteins known to stimulate cell proliferation in trans, such as AREG, NTS, EREG, TGFBI, IGF2, and IL33.

**Fig. 2 | POU5F1B enhances CRC progression in vitro and in vivo. a** Western blot analysis of HA-tagged GFP and POU5F1B in SW480 cells (representative blot out of three independent experiments). **b** Colony formation assay in GFP- and POU5F1B-overexpressing SW480 cells (n = 3 independent experiments; P = 0.011 by two-sided t-test). **c** 3-(4,5-dimethylthiazol-2-yl)−2,5-diphenyltetrazolium bromide (MTT) assay in GFP- and POU5F1B-overexpressing SW480 cells (n = 3 independent experiments; P = 4.67e-05 by two-sided Wilcoxon test). **d** Tumor volume and weight in subcutaneous implanted POU5F1B-overexpressing SW480 cells (n = 7 tumors per group; P = 0.041 by two-sided Wilcoxon test; P = 0.048 by two-sided t-test, respectively), with representative macroscopy. **e** Tumor invasion based on bioluminescence in orthotopically implanted animals (n = 10 animals per group, P = 0.038 by two-sided Wilcoxon test). **f** Number of macrometastases in liver left lobe and of additional organs affected by metastases (n = 10 animals per group, P = 0.002, P = 0.009, respectively, by two-sided t-test). **g** Number of macrometastases in liver left lobe following intrasplenic injection and splenectomy, with at right macroscopic view and microscopy of H&E-stained slices (n = 7 animals per group, P = 2.4e−04 by two-sided t-test). **h** MTT assay (n = 3 independent experiments; sh3 P = 2e−04, sh5 P = 1.19e−06 by Wilcoxon test) of LS1034 cells transduced with two different TcGT-targeting shRNAs or a control scramble sequence. **i** Tumor volume and weight in mice subcutaneously implanted with POU5F1B-downregulated or control LS1034 cells (n = 16 tumors scramble and n = 8 tumors shRNA, P = 0.018, P = 1e−04 respectively, by two-sided Wilcoxon test, boxplots left panel are represented as first and third quartiles with median in the center and whiskers are defined as 1.5 times the interquartile range clipped to the maximum/minimum of the data), with representative macroscopy (shRNA3 left, shRNA5 right). Data in **b**–**h**, **i** right) are presented as mean values ± s.e.m., with single values as circles. *P < 0.05, **P < 0.01, ***P < 0.001. Source data are provided as a Source Data file.

Next, we characterized the protein interactome of POU5F1B by affinity purification and mass spectrometry (AP-MS). Sixty proteins were found highly enriched in POU5F1B-specific immuno-precipitates (Fig. 5a and Supplementary Data 2), several known to associate in complexes endowed with signaling functions (Fig. 5a, blue balls) and others related to cytoskeleton (Fig. 5a, green balls). POU5F1B notably recruited the receptor tyrosine kinase (RTK) ERBB2, an interaction confirmed by co-immunoprecipitation in SW480, HT29, and LS1034 cells (Supplementary Fig. 5c), and six of its previously identified interactors: MST1R, another RTK; RIN1, a RAS effector protein; SORL1, a regulator of ERBB2 subcellular distribution; DNAJA3, a ubiquitin ligase chaperone that plays a role in ERBB2 degradation[30]; and POLR1A and B, two RNA polymerase I subunits that, upon nuclear ERBB2 binding, regulate rRNA synthesis and protein translation[31]. POU5F1B was also associated with other signal transduction regulators such as CEMIP, a cell migration-inducing protein previously implicated in the Wnt/βcatenin/Snail signaling pathway in CRC[32], as well as PRKCA[33], LPCAT4[34], PLEKHA5[35], TTI1[36], FKBP5[37], and TSC2[38]. Noteworthy, 14 interactions between 20 of our POU5F1B preys had been previously documented[39] (Fig. 5a, red lines).

We went on to document POU5F1B-induced changes in the cell proteome by conducting SILAC (stable isotope labeling by amino acids in cell culture) coupled to LC-MS/MS and computational analysis on lysates and supernatants of two independent pairs of GFP- and POU5F1B-overexpressing SW480 and of control and POU5F1B-knockdown LS1034 cell lines (Fig. 5b–e). Based on stringent filtering criteria (protein detected and fold-change in the same direction in both experiments; high-intensity candidates –over quantile 25%-; and p value < 0.05), we found the levels of respectively 54 and 167 proteins to be increased in lysates of POU5F1B-overexpressing SW480 or scramble shRNA-transduced LS1034 cells compared with their respective controls (Fig. 5b, c and Supplementary Data 3 and 4). There was a strong overlap in GO terms obtained by subjecting these two sets of proteins to the DAVID 6.8 annotation tool, including extracellular exosome, focal adhesion, cytoskeleton, or membrane substructures such as caveolae, desmosomes, rafts, and ruffles, that is, processes linked notably to cell growth, cell migration and cell-to-cell communication. Of note, this enrichment for membrane structures correlates with our finding of POU5F1B-enriched and possibly altered DRMs (Fig. 4d). Levels of SerpinA1, a serine protease inhibitor previously reported to promote CRC progression[40], were increased in the presence of POU5F1B in both SW480 and LS1043 cells, as were those of ROCK2 and its target EZR, two kinases regulating cytoskeletal reorganization, cell adhesion, and cell motility[41,42]. Interestingly CEMIP, a cell migration-inducing protein[32] and one of the strongest interactors identified in our POU5F1B pull-down assay, was expressed at significantly higher levels in control than POU5F1B-depleted LS1034 cells, as were IL33, also previously reported to facilitate CRC cell proliferation[43], and CEA-CAM5 (CEA), a cancer biomarker discovered more than 50 years ago[44] and since then identified as playing a role in CRC progression and metastasis[45].

Our SILAC-based comparison of the supernatants of POU5F1B-high and -low SW480 and LS1034 cells lighted up 27 and 73 proteins POU5F1B-upregulated proteins, respectively (Fig. 5e, d and Supplementary Data 5 and 6). They included molecules known to act as ligands of transmembrane receptors (JAG1, AREG, ESM1, ADM, MST1, and IL1RN). Correlating these data, conditioned medium from POU5F1B-overexpressing SW480 cells stimulated the colony formation potential of their POU5F1B-negative controls (Fig. 5f). POU5F1B upregulated secreted factors also included cell adhesion-related proteins such as TNC, an extracellular matrix component predictive of liver metastasis in CRC[46], as well as SPOCK2, IGFBP7, SEMA7A, FBLN1, LTBP2, DSG1, FBLN7, and TLN1.

To complement this study, we asked whether POU5F1B modified the phospho-proteome of CRC cells. For this, we compared lysates from two independent pairs of GFP- and POU5F1B-overexpressing SW480 cell lines using an array of 1325 antibodies (Kinex™), 450 of which recognized proteins irrespectively of their post-translational status and 875 only their phosphorylated subset. Twenty-eight proteins were found enriched (p value < 0.05, spot intensity >1000, and POU5F1B vs. GFP fold change >1.2) in POU5F1B-overexpressing SW480 cells (Supplementary Fig. 5d and Supplementary Data 7). Ten of them were detected with the corresponding pan-specific antibodies and 18 with their phospho-specific counterparts. Standing out were 5 protein kinases linked to cytosolic signal transduction upon transmembrane receptor activation (PRKCD, PRKCG, PI3KCA, PIK3R2, and PDPK1), 2 cyclin-dependent kinases (CDK14 and CDK15) and 13 proteins previously linked to cytoskeletal rearrangements (CFL2, WNK1, PRKCD, RPS6KB1, PFN1, Pim3, PRKCM, PAK1, DYRK2, PIK3CA, PAK2, DMPK1, and CTNNB1).

## POU5F1B-encoding TcGTs are detected in several other cancers

To probe for a broader role of POU5F1B in human cancer, we sought the presence of *POU5F1B* TcGTs in more than 9,500 samples from 32 tumor types represented in the TCGA RNA-seq dataset. The *LTR66-POU5F1B* TcGT predominated in CRC, while shorter versions starting at downstream TEs seemed more frequent in other cancers (Fig. 6a, left). While insufficient processivity of the RT reaction may have led to an underestimation of the 5' extension of lowly expressed transcripts in some cases, publicly available PACBIO and CAGE data from a gastric and a breast cancer cell line revealed that POU5F1B-encoding TcGTs were produced from *LTR66* in the former and from downstream *Alu* or *LTR33* integrants in the latter (Supplementary Fig. 6a). *POU5F1B* TcGTs generally correlating with a higher level of *POU5F1B* expression were most frequent in CRC samples, but also commonly detected in several other tumors, including prostate, uterus, breast, lung and stomach cancers (Fig. 6a, right). Similar results were obtained from a collection of more than 600 cancer cell lines (EGAD00001000725[18]) (Supplementary Fig. 6b). In contrast,

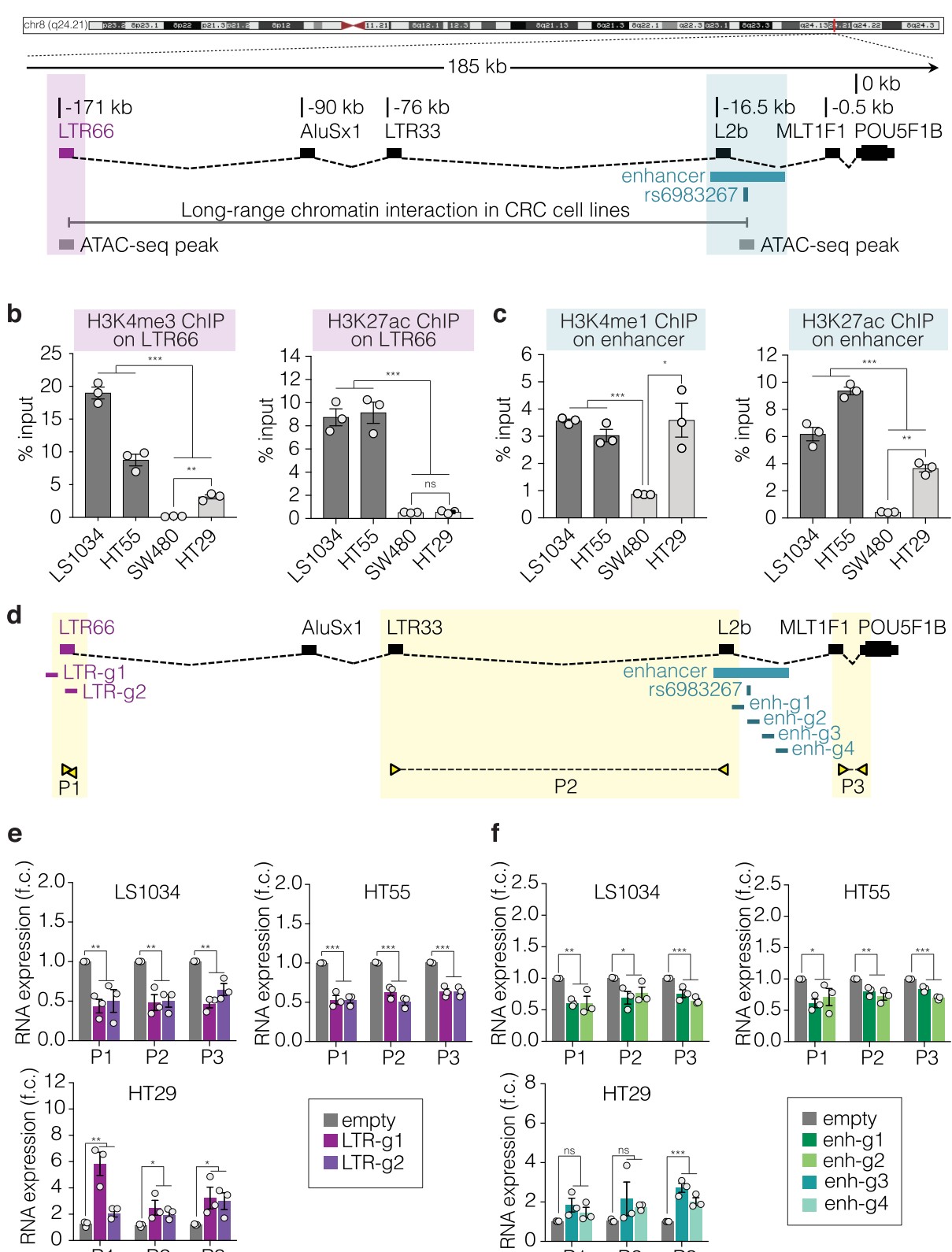

POU5F1B was lowly expressed and POU5F1B TcGTs were undetectable in more than 8,800 samples from 29 different normal tissues of the Genotype–Tissue Expression (GTEx) dataset[47,48] (Fig. 6a). POU5F1B RNA levels were similarly low in human embryonic stem cells (hESC), at least 1,000-fold below those of its paralog OCT4 (Supplementary Fig. 6c). In all tumors where high levels of POU5F1B transcripts were documented, they originated from some upstream TE, POU5F1B RNAs devoid of sequences 5′ of the annotated TSS being only exceptionally detected. Together, these data demonstrate that the tumor-preferential overexpression of POU5F1B through the onco-exaptation of normally silenced TE promoters is a widespread phenomenon in human cancer.

**Fig. 3 | The rs6983267-containing enhancer regulates LTR66-POU5F1B expression. a** The LTR66-POU5F1B genomic locus, with depiction of TcGT (black boxes linked by broken dashed lines), rs6983267-containing enhancer from UCSC GeneHancer tracks (turquoise box), long-range chromatin interaction in CRC cell lines from the literature[25] (gray line), and ATAC-seq peaks in CRC tumor samples from TCGA[26] (gray boxes). **b, c** ChIP-PCR analyzes of LTR66-POU5F1B-expressing (LS1034 and HT55) and non-expressing (SW480 and HT29) CRC cell lines for indicated chromatin marks (*n* = 3 independent experiments; from left to right **b**: ***P* = 6.70e−08, ***P* = 9.64e−03, ****P* = 5.58e−07, ns*P* = 0.68; **c**: ****P* = 6.45e−04, **P* = 4.81e−02, ****P* = 1.01e−05, ***P* = 6.38e−03, respectively, by two-sided *t*-test). **d** Schematic representation of gRNAs (g) and primers (P) used to target CRISPRi or

CRISPRa to either LTR66 or the rs6983267-containing region and measure LTR66-POU5F1B transcripts by qRT-PCR. **e,f** Impact of indicated manipulations on LTR66-POU5F1B RNA levels (*n* = 3 independent experiments; **e**: LS1034 P1 ***P* = 4.21e−03, P2 ***P* = 1.33e−03, P3 ***P* = 4.67e−03; HT55 P1 ****P* = 6.79e−05, P2 ****P* = 6.03e−05, P3 ****P* = 7e−05; HT29 P1 ***P* = 6.30e−03, P2 **P* = 4.93e−02, P3 **P* = 4.32e−02; **f**: LS1034 P1 ***P* = 2.96e−03, P2 **P* = 2.94e−02, P3 ****P* = 9.62e−03; HT55 P1 **P* = 2.03e−02, P2 ***P* = 3.92e−03, P3 ****P* = 3.69e−05; HT29 P1 ns *P* = 8.64e−02, P2 ns*P* = 0.18, P3 ****P* = 6.62e−04, respectively, by two-sided *t*-test). Data in **b**–**f** are presented as mean ± s.e.m., with single values as circles. Ns, not significant; **P* < 0.05, ***P* < 0.01, ****P* < 0.001. Source data are provided as a Source Data file.

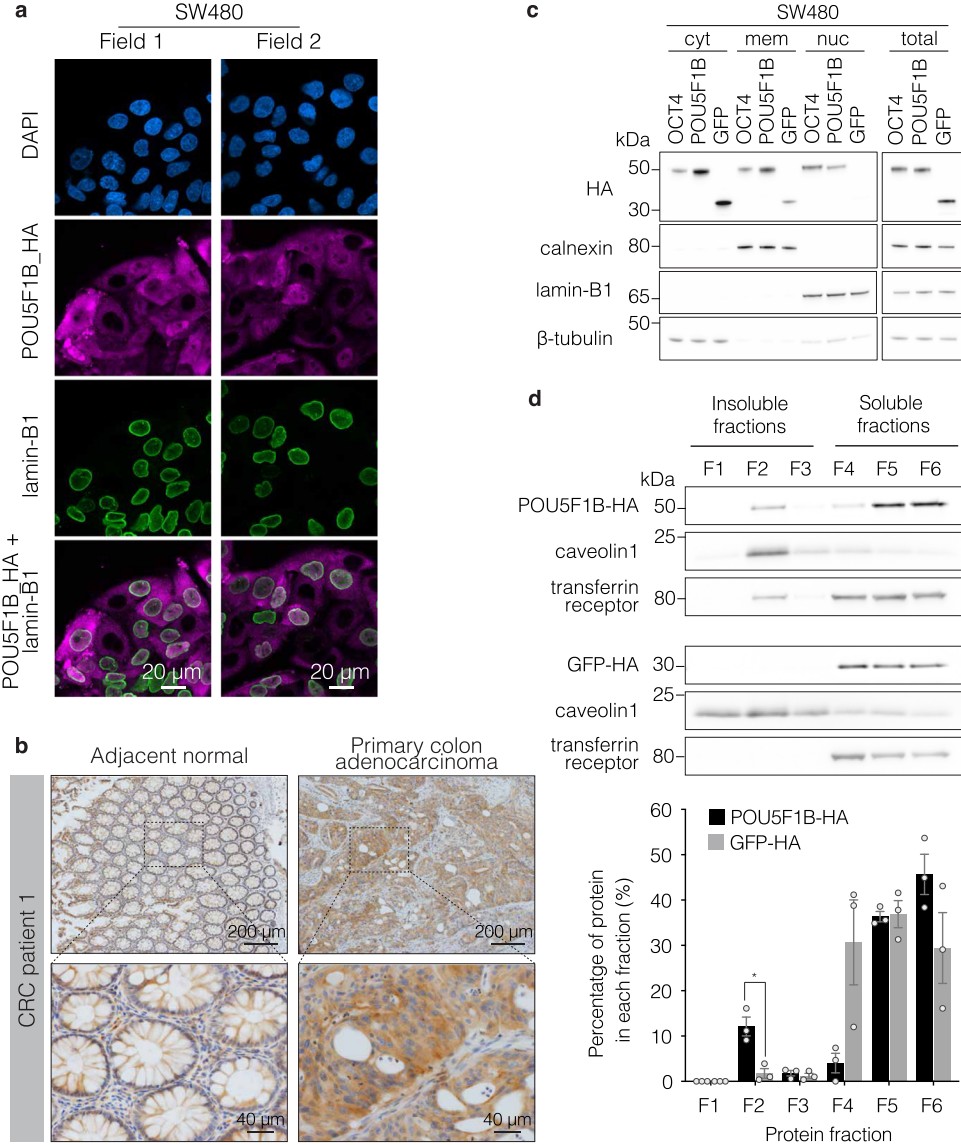

**Fig. 4 | POU5F1B differs from OCT4. a** Representative immunofluorescence–confocal microscopy of POU5F1B-HA overexpressing SW480 cells, with DAPI in blue, HA in pink, and the nuclear membrane marker lamin-B1 in green. Two fields out of eight are shown. **b** Immunohistochemistry for endogenous POU5F1B in adjacent normal colon and primary colon adenocarcinoma samples from CRC patient 1. Two levels of magnification from a representative field out of three are shown. More patients are depicted in Supplementary Fig 4e. **c** Subcellular fractionation into the cytoplasm (cyto), membrane (mem), and nuclear (nuc) compartments from OCT4-, POU5F1B- and GFP-overexpressing SW480 cells, with total cell extracts on right. Calnexin, lamin-B1, and beta-tubulin

are used as controls for membrane, nucleus, and cytoplasm, respectively (representative blot out of three independent experiments). **d** Top, isolation of detergent-resistant membranes (DRMs) from POU5F1B- and GFP-overexpressing SW480 cells. F1–F3 correspond to insoluble fractions, F4–F6 to soluble fractions, DRMs being traditionally found in fraction F2. Caveolin1 and transferrin receptors are used as controls for insoluble and soluble fractions, respectively (representative blot out of three independent experiments). Bottom, western blot quantification indicating the percentage of protein in each fraction (*n* = 3 independent experiments; *P* = 2.1e−02 by two-sided *t*-test). Source data are provided as a Source Data file.

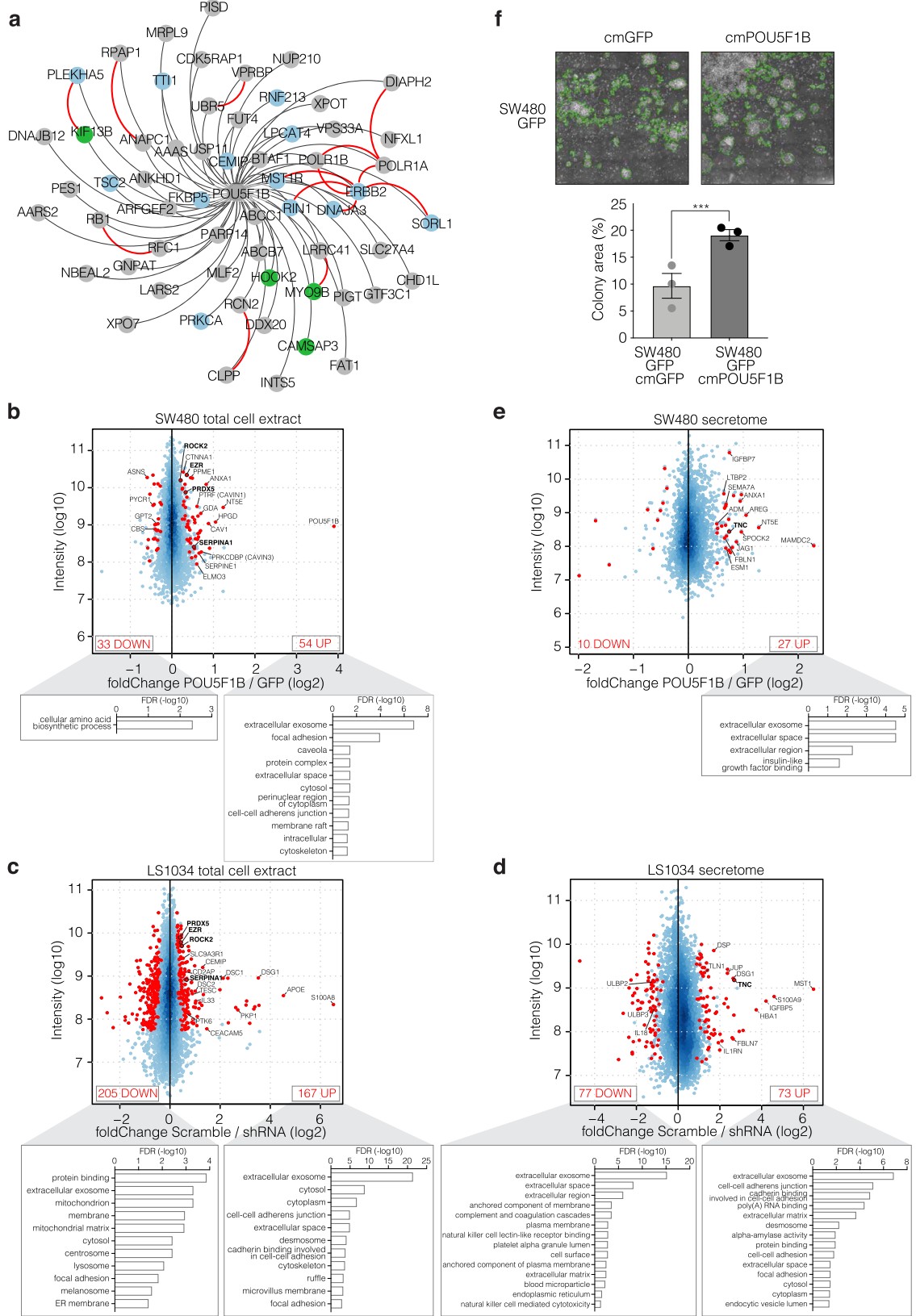

## Discussion

The *POU5F1B* retrogene arose by retrotransposition of *OCT4* in the last common ancestor of great apes, hence is absent in the mouse, the animal model most commonly used to study human cancers. Furthermore, POU5F1B displays biological properties fundamentally different from those of its highly conserved OCT4 relative. As such,

POU5F1B is a dually novel oncogene. The chromosomal region between *LTR66* and *POU5F1B* hosts cancer risk loci for breast, colon, and prostate cancer[49]. Here, we demonstrate that production of the *LTR66-POU5F1B* TcGT is stimulated by an intronic enhancer previously found to display a state of open chromatin in several human cancers, including colon, breast, and prostate[26]. This sequence was initially

**Fig. 5 | Proteomic characterization of POU5F1B-induced molecular changes.**
**a** High-confidence interactome of POU5F1B detected by AP/MS in HT29 and LS1034 CRC cells overexpressing POU5F1B-HA. Weighted black edges are based on the average fold change over controls ($n = 12$ samples, 4 conditions, 3 replicates/condition; all depicted interactions have a fold change over controls >5 and $P < 0.01$). Red edges depict previously documented protein-protein interactions. Interactors with signaling functions are highlighted in blue and cytoskeleton-related proteins in green. **b**–**e** MA plot depicting relative abundance and average intensity of individual proteins identified by SILAC in total cell extracts (**b**, **c**) or in the secretome (**d**, **e**) of POU5F1B- vs. GFP-overexpressing SW480 cells (**b**, **e**) or in sh-scramble vs. shRNA3 & shRNA5 LS1034 cells (**c**, **d**). All SILAC measurements were performed in independent duplicates, each dot represents a detected protein, with significantly changed ones ($P < 0.05$, outlier detection test as computed by MaxQuant) in red and their numbers indicated in upper corners. Names in bold are for proteins common to the corresponding settings in both cell lines. Significantly enriched proteins were clustered in GO terms using the Functional Annotation Chart from DAVID bioinformatics resources 6.8. **f** Conditioned-medium colony formation assay; GFP-overexpressing SW480 cells form bigger colonies when exposed to POU5F1B- rather than GFP-conditioned medium (cm). A green line delimits the quantified area in representative pictures ($n = 3$ independent experiments with 3 replicates; $P = 1.27e{-}05$ by two-sided $t$-test). Data presented as mean ± s.e.m., with single values as circles. Source data are provided as a Source Data file.

identified as a *MYC* enhancer, but evidence indicates that it also acts on the *LTR66* integrant situated upstream, including i) a physical interaction between the two elements documented in several CRC cell lines[25], ii) a concordance of their ATAC-seq profiles in CRC tumors from the TCGA cohort[26], and iii) our functional data. Of note, we found that the production of *LTR66-POU5F1B* TcGTs is not linked to a polymorphism previously mapped to this enhancer and also that it can be bi-allelic. Thus, induction of *LTR66-POU5F1B* might result from *trans*-acting influences or from *cis*-acting mutations situated at greater distances and affecting both alleles. However, the finding that in other cancers, POU5F1B-encoding TcGTs can arise from TE integrants situated downstream of *LTR66* argues for a strong selective pressure for expression of this protein irrespective of the underlying mechanism.

We found POU5F1B overexpression to be a negative CRC prognostic marker in the SYSCOL cohort. This corroborates similar observations in hepatocellular carcinoma and gastric cancer[50,51]. Furthermore, POU5F1B-encoding transcripts have been detected in circulating but not primary tumor cells from pancreatic ductal adenocarcinoma (PDAC), where they have been found to be associated with a more rapid clinical deterioration[52], suggesting that POU5F1B contributes to conferring PDAC cells with the phenotype of circulating and, ultimately, of metastasis-initiating cells (MICs)[53]. Accordingly, the apparent absence of prognostic value for POU5F1B in the TCGA dataset is intriguing and could stem from differences in patient populations and therapeutic regimens compared with the SYSCOL cohort.

Our protein-centered analyzes suggest a prominent role in the activation of intracellular signaling events and cytoskeletal rearrangements in the pro-oncogenic effects of POU5F1B. First, the retrogene product could be co-immunoprecipitated in CRC cells with a number of signaling molecules, including protein kinases such as MST1R, PRKCA, ERBB2 and several of its known interactors, as well as modulators of GTPase activation such as the RAS effector protein RIN1. Second, POU5F1B expression induced the upregulation or increased phosphorylation of several downstream mediators of protein kinase–mediated signaling. Third, POU5F1B was enriched in sphingolipid/cholesterol-enriched membrane subdomains or lipid rafts, where signaling effectors are commonly concentrated and was associated with increased levels of several components of caveolae, which are biochemically closely related to lipid rafts and play an important role in the trafficking of signaling receptors such as ERBB2, EGFR, insulin-R, TGFR and PDGFR[54–56]. Caveolae are also associated with the activation of the contractile actin cytoskeleton, which favors cell migration[57], and we relatedly found POU5F1B to interact with or to induce the upregulation and/or phosphorylation of several mediators of cytoskeletal rearrangements. This suggests that modifications of cell architecture and motility may underlie the prometastatic effect of POU5F1B documented in our xenotransplantation experiments.

Our results have important medical implications. First, they suggest that the detection of POU5F1B RNA or protein in a CRC biopsy warrants aggressive management of the underlying tumor. Second, although a physiological role for the recently emerged *POU5F1B* is yet to be identified, this gene is highly tolerant to loss-of-function mutations (pLI = 0), and can be bi-allelically inactivated in some individuals (https://gnomad.broadinstitute.org/)[58], and displays little expression in normal tissues. Accordingly, its product is an attractive target for the development of novel cancer therapies. Third, blocking the *trans*-acting mediators released by POU5F1B-expressing cells could also be of benefit, as these molecules are predicted to increase the oncogenic properties of neighboring cells and of the tumor microenvironment. Finally, the presence of *POU5F1B*-encoding TcGTs in several other malignancies, including breast, prostate, stomach, and uterus, the previous identification of the *POU5F1B* locus as an integration hotspot for human papillomavirus and the coincidence of its transcripts with more advanced histological grades in cervical cancer[59,60], as well as the documentation of its amplification in some gastric tumors[51] and growth-promoting effect on hepatocellular carcinoma cell lines[50] all indicate that our findings have a relevance that likely extends well beyond colorectal cancer alone.

## Methods
### RNA sequencing
Reads were mapped to the human genome (hg19) using hisat2 (2.1.0)[61]. Samtools (1.4) were used to manipulate the alignments. Counts on genes and TEs were generated using featureCounts (from subread 1.5.2)[62]. To avoid read assignation ambiguity between genes and TEs, a gtf file containing both was provided to featureCounts. For repetitive sequences, an in-house curated version of the RepeatMasker database (Smit, AFA, Hubley, R & Green, P. RepeatMasker Open-4.0. 2013–2015; http://www.repeatmasker.org) was used, where fragmented LTR elements were fused together and flanking LTR elements were grouped with their internal counterparts. Only uniquely mapped reads were used for counting on genes and TEs. Library sizes used in the CPMs' calculation were computed with the TMM method as implemented in the limma package of Bioconductor[63]. Library sizes for genes were used for both genes and TEs. Differential gene expression analysis was performed as explained in the 'David enrichment analysis' methods chapter.

### Transpochimeric gene transcripts analysis
First, a per sample transcriptome was computed from the RNA-seq bam file using Stringtie (1.3.4c)[64] with parameters –j 1 –c 1. For TCGA, samples were downsampled down to 17mio reads in order to match the average sequencing depth of the SYSCOL dataset. Each transcriptome was then crossed using BEDTools (2.30.0)[65] to both the ensemble hg19 coding exons and curated RepeatMasker to extract TcGTs for each sample. Second, a custom program was used to annotate and aggregate the sample level TcGTs into counts per groups (normal, tumor, tissue, or cancer type depending on the dataset). In brief, for each dataset, a gtf file containing all annotated TcGTs was created and TcGTs having their first exon overlapping an annotated gene or TSS not overlapping a TE were discarded. From this filtered file, TcGTs associated with the same gene and having a TSS 100 bp within each other were aggregated. Finally, for each aggregate, its occurrence per group was computed.

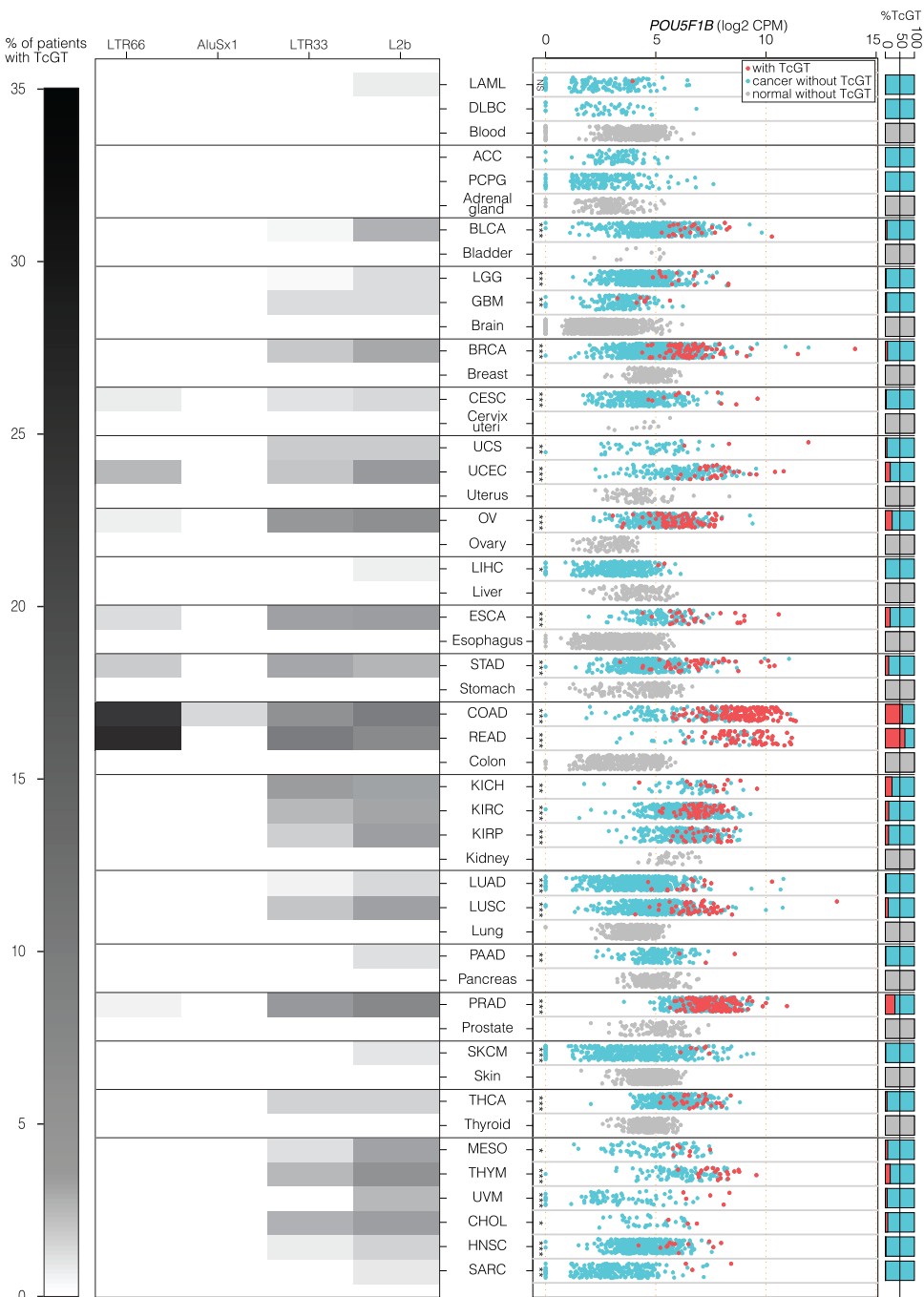

**Fig. 6 | POU5F1B-encoding TcGTs are detected in several other cancers. a** Left, heatmap depicting the percentage of patients with POU5F1B TcGTs for indicated TCGA cancers and normal tissues of the Genotype–Tissue Expression (GTEx) dataset[47]. Locations of TEs acting as TSS are indicated on top, with the intensity of gray shade proportional to usage frequency. Right, levels of expression of POU5F1B transcripts in 9,566 samples from 32 cancer types of the TCGA dataset and in 8878 samples from 29 normal tissue types of the GTEx dataset. For each TCGA and GTEX category, expression of TcGT samples was compared to non-TcGT samples with a two-sided t-test. Significance levels are shown as stars (*** pval < 0.001, **pval < 0.01, *pval < 0.05). Right margin, percentage of samples with (red) or without TcGT (blue -cancer-, gray -normal tissue–). Abbreviations: ACC adreno-cortical carcinoma; BLCA bladder urothelial carcinoma; BRCA breast carcinoma;

CESC cervical squamous cell carcinoma; CHOL cholangiocarcinoma; COAD colon adenocarcinoma; DLBC diffuse large B-cell lymphoma; ESCA esophageal carci-noma; GBM glioblastoma multiforme; HNSC head, and neck squamous cell carci-noma; KICH kidney chromophobe; KIRC kidney renal clear cell carcinoma; KIRP kidney renal papillary cell carcinoma; LAML acute myeloid leukemia; LGG low grade glioma; LIHC liver hepatocellular carcinoma; LUAD lung adenocarcinoma; LUSC lung squamous cell carcinoma; MESO mesothelioma; OV ovarian cancer; PAAD pancreatic adenocarcinoma; PCPG pheochromocytoma, and paraganglioma; PRAD prostate adenocarcinoma; READ rectal adenocarcinoma; SARC sarcoma; SKCM skin cutaneous melanoma; STAD stomach adenocarcinoma; THCA thyroid carci-noma; THYM thymoma; UCEC uterine corpus endometrial carcinoma; UCS uterine carcinosarcoma; UVM uveal melanoma.

## Analysis of public datasets

Raw RNA sequencing reads were downloaded from NCBI's dbGaP for both the TCGA and the GTEx[47] datasets. Raw single-cell RNA-seq data[17] was downloaded from the European Genome–phenome Archive (EGAD00001002727). Raw RNA-seq data from cancer cell lines were downloaded from the EGAD00001000725 repository. RNA-seq data from eighteen CRC patients were downloaded from GEO (GSE50760). RNA-seq data from H1 and H9 cell lines and the collection of hESC clones were downloaded from GEO (GSE60945, GSE83765)[66,67]. Long-range chromatin interactions in three CRC cell lines were obtained from Jäger, R. et al.[25]. TCGA ATAC-seq peaks were downloaded from https://gdc.cancer.gov/about-data/publications/ATACseq-AWG.

## CMS classification

The CMS classification was obtained as implemented in the CMSclassifier software (https://github.com/dFenix7/CMSclassifier).

## 5′-rapid amplification of the cDNA ends (5′-RACE)

We applied SMARTer 5′-RACE technique in LS1034, HCT116, LS174T, NCI-H508 and LoVo CRC cell lines, according to the manufacturer's protocol (Clontech) using the gene–specific primer:

5′ GATTACGCCAAGCTTGCCACAAACTCCAGGTTCTCTTTCCCT AGCTCCTC 3′. 5′-RACE products were loaded in an agarose gel and a fragment around 2,500 bp was excised for DNA purification and In-Fusion cloning (Clontech). Ten transformed colonies per condition were analyzed by restriction digestion and only those with the RACE insert were Sanger sequenced. A scheme of the sequenced mRNA species and their frequency is shown.

## Cell culture

LS1034, NCI-H508, and DLD1 cell lines were obtained from the American Type Culture Collection (ATCC) and maintained in RPMI 1640 medium (Gibco) supplemented with 10% FCS (Bioconcept 2-01F36-I); LS174T (ATCC) and HT55 (Sigma) cells were cultured in EMEM supplemented with 10% FBS, HT55 being complemented with 2 mM glutamine (Gibco) and 1% of non-essential aminoacids (Sigma). SW480 and SW620 (ATCC) were cultured in L15 medium (Sigma); HT29 (ATCC) in McCoy's 5 A (Thermo Fisher); LoVo (ATCC) in Ham's F12K (ThermoFisher); and HCT116 (ATCC) in DMEM (Gibco), being all supplemented with 10% FCS. For lentiviral vector production, 293 T cells were cultured in DMEM supplemented with 10% FBS with 100 IU ml⁻¹ penicillin, 100 ug ml⁻¹ streptomycin, and 26 μg ml⁻¹ glutamine (Corning 30-009-CI) at 37 °C in a humidified atmosphere of 5% $CO_2$. All cells tested negative for mycoplasma.

## Overexpression and shRNA vectors

For the POU5F1B-expressing vector, genomic DNA from DLD1 cells (ATCC® CCL-221™) was PCR amplified with the POU5F1B primers 5′-CACCATGGCGGGACACCTGGCTTCGGATTTC-3′, 5′- GTTTGAATGCAT GGGAGAGCCCAG-3′, cloned into a pENTR TOPO donor vector (Thermo Fisher), that was recombined with the doxycycline–inducible lentiviral destination vector pSin-TRe–3xHA-puro. Short hairpin RNA (shRNA) were designed with the help of the i-Score Designer website, and adapted to the miR-E shRNA structure and cloned into the LT3GEPIR pRRL backbone following Fellmann C et. al. instructions[68]: shRNA3 5′- TGCTGTTGACAGTGAGCG**CACAGGT GATTATGATTTAAAG**TAGTGAAGCCACAGATGTA**CTTTAAATCATA ATCACCTGTG**TGCCTACTGCCTCGGA-3′; shRNA5 5′- TGCTGTTGA CAGTGAGCG**CACATTCAGTCAACATTTAATG**TAGTGAAGCCACAGA TGTA**CATTAAATGTTGACTGAATGTG**TGCCTACTGCCTCGGA-3′) with a scrambled shRNA serving as negative control 5′- TGCTG TTGACAGTGAGCG**CAACAAGATGAAGAGCACCAAG**TAGTGAAG CCACAGATGTA**CTTGGTGCTCTTCATCTTGTTG**TGCCTACTGC CTCGGA-3′. The luciferase–expressing vector pCDH-EF1-Luc2-P2A-tdTomato (Addgene) was transduced in cells used for xenotransplantation experiments.

## Lentivirus production and stable gene expression

Lentiviral particles produced as described at http://tronolab.epfl.ch were used to transduce SW480, HT29, SW620, and LS1034 cells in 6-well plates at multiplicity of infection 1, before selection in 1 μg ml⁻¹ puromycin for 7 days. HT55 and LS174T cells were transduced by spinoculation at 600 g at 32 degrees for 60 minutes with 8 ug mL⁻¹ polybrene (Sigma) at a multiplicity of infection 100 before selection in 1 μg ml⁻¹ puromycin for 7 days. POU5F1B and GFP overexpression in SW480, HT29, and SW620 was verified, after three days of 500 ng ml⁻¹ doxycycline activation, by western blot with HRP-conjugated anti-HA antibody (clone 3F10 Roche 12013819001, 1:1,000); and *POU5F1B* TcGT downregulation upon shRNA transduction in LS1034, HT55, and LS174T was analyzed by quantitative real-time PCR using P3 pair of primers.

## In vitro proliferation and colony-forming assays

Cell proliferation was determined by the 3-(4,5-dimethylthiazol-2-yl) −2,5-diphenyltetrazolium bromide (MTT) assay for 5–6 days, staining the cells with 5 mg ml⁻¹ MTT for 3 h, removing the medium, adding DMSO to dissolve the crystals and measuring absorbance at 560 nm. Colony-formation assay (CFA) was performed by seeding 400 cells (HT29, SW480 and SW620) in 6-well plates, stained with crystal violet after 2 weeks of growth, and analyzed with Fiji software. Similarly, soft agar CFA was performed by seeding 10,000 cells (LS1034) and staining the colonies with MTT. 500 ng ml⁻¹ doxycycline–containing medium was refreshed every 3 days until the end of each experiment. For the conditioned medium CFA, SW480 GFP-expressing cells were seeded with a mixture of 2/3 regular culture medium and 1/3 conditioned medium, previously harvested from GFP- or POU5F1B-expressing SW480 cells, passed through 0.22 μm filter, aliquoted and stored at −80 °C. The medium mix was changed every other day until the end of the 2-week experiment.

## Mouse studies

NOD/SCID/gamma mice were maintained under standard animal housing conditions in a normal 12 h light–dark cycle with ad libitum access to food and water. All animal experiments were performed within the EPFL animal facility, in accordance with the Swiss Federal Veterinary Office guidelines and as authorized by the Cantonal Veterinary Office (animal license VD3381). Human CRC cell line SW480 expressing GFP or POU5F1B under doxycycline inducible promoter and expressing luciferase (3 million cells diluted in 200 μl Matrigel BD) were grafted subcutaneously in anesthetized NOD/SCID/gamma (NSG 5557) immunodeficient mice (Charles River), each cell type in one side of the mouse back ($n = 7$). Tumor volumes were estimated each 3–4 days from two-dimensional caliper measurements using the equation V = (π/6) x L x W2, where V = volume (mm³), L = length (mm), and W = width (mm), and reported as volume mean ± s.e.m for each mouse group. Mice were sacrificed 38 days after injection and tumor weight measured. Two additional subcutaneously implanted animals were used as tumor donor for the orthotopic model. Once the suitable tumor size was reached (600–800 mm³) it was cut into 2 × 2 mm³ pieces and implanted onto the cecum wall of 10 animals as previously described[69] ($n = 10$). Bioluminescence imaging was performed on an IVIS Spectrum In Vivo Imaging System (Perkin Elmer) and analyzed with Living Image Software (Perkin Elmer). For the metastasis model without the influence of tumor burden, 1.5e06 SW480 or SW620 cells modified as specified above were injected into the spleen of 14 mice (7 mice for each condition) or 18 mice (9 mice for each condition), respectively. To avoid local tumor growth, spleens were removed by cauterization 5 min after cell injection. Mice were sacrificed 5 weeks later, and hepatic metastases were examined macroscopically and

microscopically following H&E tissue staining. Mice were maintained on doxycycline food pellets (0.625 g kg⁻¹; SAFE E8404 version 0002). Subcutaneous and intrasplenic injection experiments with LS1034 cells were performed in 3 groups (sh3, sh5, scramble) of 8 animals each, which were fed with doxycycline food pellets from one week before injection to the experimental endpoint. LS1034 cells were treated with doxycycline 72 h before injection.

## Micrometastases quantification

Images were acquired from Hematoxylin&Eosin stains on an Olympus VS120 Whole Slide Scanner, using a 20x objective (UPLSAPO, N.A. 0.75) and a color camera (Pike F505 Color) with an image pixel size of 0.345 microns. Obtained images were analyzed using the software QuPath (0.3.2)[70] using groovy scripts, making use of a pixel classifier to segment and measure cancer cell clusters. For more details, please access the following Zenodo project (https://doi.org/10.5281/zenodo.6523649).

## Chromatin immunoprecipitation

Chromatin was prepared as described previously[5]. Pellets were lysed by resuspension in LB1 for 10 min (50 mM HEPES-KOH pH 7.4, 140 mM NaCl, 1 mM EDTA, 0.5 mM EGTA, 10% Glycerol, 0.5% NP40, 0.25% Tx100, protease inhibitors), centrifuged, resuspended in LB2 for 10 min (10 mM Tris pH 8.0, 200 mM NaCl, 1 mM EDTA, 0.5 mM EGTA and protease inhibitors), centrifuged, and resuspended in LB3 (10 mM Tris pH 8.0, 200 mM NaCl, 1 mM EDTA, 0.5 mM EGTA, 0.1% NaDOC, 0.1% SDS and protease inhibitors). LB incubations were done in a rotating wheel at 4 °C and centrifugations at 1,700 g for 5 min at 4 °C. Cells in LB3 were sonicated (Covaris e220 settings: 5% duty factor, 200 cycles per burst, 140 PIP, 20 min, 4 °C) yielding genomic DNA fragments with a bulk size of 100–300 bp. Beads were coated with antibodies specific for H3K4me3 (Cell Signaling 9751), H3K27ac (Abcam 4729), and H3K4me1 (Diagenode 037–050) for 4 h at 4 °C and incubated with chromatin overnight at 4 °C before two serial washes in low salt buffer (10 mM Tris pH 8, 1 mM EDTA, 150 mM NaCl, 0.15% SDS), one in high salt buffer (10 mM Tris pH 8, 1 mM EDTA, 500 mM NaCl, 0.15% SDS), one in LiCl buffer (10 mM Tris pH 8, 1 mM EDTA, 0.5 mM EGTA, 250 mM LiCl, 1% NP40, 1% NaDOC), and one in TE buffer. Immunoprecipitated DNA was purified (MinuElute PCR purification kit Qiagen), and up to 10 ng of immunoprecipitated or input DNA was used for quantitative PCR analysis using P1 primers (5′-GGCCTCA-CACCGAATAACTC-3′, 5′-ACGAGGAGCAGTCTCCTGAA-3′) for *LTR66*; 5′-ACAGAGCATTCCCACTGGAC-3′, 5′-TGGCGAGACAATACTTGCAG-3′ for the enhancer.

## CRISPRi and CRISPRa experiments

sgRNA were designed with CRISPOR software[71] and cloned into a pLenti-SpBsmBI-sgRNA-Hygro vector (Addgene 62205). Two and four sgRNA were used to target the LTR66 and enhancer regions, respectively:

LTR-g1 5′- TCACATCATTCTCACCACTCTGG-3′;
LTR-g2 5′- GGAGCAGTCTCCTGAAGCTTTGG-3′
enh-g1 5′-GACGATGAGGGTATTAACTCTGG-3′;
enh-g2 5′- GGTAATATGTTTGGGCCTGTAGG-3′;
enh-g3 5′- CTTGCTGGTAGAACTTACGTAGG-3′;
enh-g4 5′- TGCATTGATATAGGCCAAACTGG-3′.

LS1034 and HT55 cells were transduced with dCas9-KRAB lentiviral vector (Addgene 71236), and HT29 cells with dCas9-VPR lentiviral vector (Addgene 99373) and selected in 1 µg ml⁻¹ puromycin for 7 days before transduction with sgRNA-producing lentiviral vectors and selection in 200 µg ml⁻¹ hygromycin (Roche 10843555001) for 7 days. Expression levels of *POU5F1B* TcGT were checked by quantitative real-time PCR using primers P1, P2 (5′-GAGCCAAGAGAA GACGTCCAG-3′, 5′-GCTTGTGACTTAGCCTGGGTG-3′); and P3 (5′- CT GAGGCCTCCTACCAACAG-3′, 5′- TGCTATGGAATGGTGTGTCC-3′).

## Evolutionary tree of POU5F1B paralogs

We built an evolutionary tree by using syntenic regions of *POU5F1B* closest paralog (>80% homology with human POU5F1B according to BLAT) inferred in indicated species using UCSC liftOver tool (with option minMatch = 0.5, with following genome versions: hg19, panTro5, gorGor4, ponAbe2, nomLeu3, rheMac8, macFas5, calJac3, and tarSyr2). We built the tree by merging branches whenever two paralogs had the same genomic environment in different species. *MYC* was used as an anchor gene to double–check its distance from each *POU5F1B* paralog.

## Immunofluorescence

Cell lines were plated on glass coverslips into 24-well plates and cultured for 3 days up to 70% confluence in 500 ng ml⁻¹ doxycycline–containing medium. Growth medium was refreshed (1 ml) prior to a 15 min fixation with 1 ml 8% paraformaldehyde (PFA) added dropwise. Cells were washed three times with PBS, permeabilized with 0.1% saponin PBS for 20 min, and blocked with 1% BSA 0.1% saponin PBS for 30 min before incubation with anti-HA (bioLegend, Covance catalog #MMS-101P, 1:1,000) and anti-lamin-B1 (abcam ab16048, 1:1,000) antibodies in 1% BSA 0.1% saponin PBS 2 h at room temperature (RT) under agitation. Samples were washed three times with PBS and incubated with Alexa 647-conjugated (A647) anti-mouse antibody (1:1,000) and A568 anti-rabbit antibody (1:1,000) in 1% BSA 0.1% saponin PBS for 40 min at RT. Three final washes were performed before mounting the slides in Vectashield with DAPI (Vector Laboratories). Images were acquired on a ZEISS LSM 700 confocal microscopy and analyzed with Fiji software.

## Immunohistochemistry

Detection of POU5F1B was performed manually on paraffin-embedded CRC cell lines (SW480, HT29, LS1034, and HT55) and primary CRC biopsies with a rabbit polyclonal custom anti-POU5F1B antibody (1:1,000) ordered from Biotem (https://www.biotem-antibody.com/) and obtained using as immunogen a 16 aa peptide (GDGPWGAEPGWVDPLT), 3 residues of which (underlined) differ between POU5F1B and OCT4. A heat pretreatment using 0.1 M Tri-Na citrate pH6 for 20 minutes at 95 °C (PT module, Thermo scientific) was applied before incubation of the primary antibody overnight at 4 °C. After incubation of a donkey anti-rabbit HRP (Jackson Immunoresearch, diluted 1:100), revelation was performed with DAB (3,3′-Diaminobenzidine, Sigma-Aldrich). Sections were counterstained with Harris hematoxylin and permanently mounted.

## Subcellular fractionation

Twenty million SW480 GFP- and POU5F1B-expressing cells were washed three times with cold PBS and harvested upon scrapping in 0.4 ml cold buffer A (10 mM KOAc, 2 mM MgOAc, 20 mM HEPES pH 7.2, 0.5 mM DTT, 0.015% digitonin) containing protease inhibitor cocktail (Roche). After centrifuging at 300 g 4 °C for 5 min, supernatant was collected as cytoplasmic fraction and pellet was resuspended in 0.4 ml buffer B (10 mM HEPES pH 7.9, 10 mM KCl, 0.1 mM EDTA, 1 mM DTT, 0.5% Triton X100, 100 mM NaF) with protease inhibitors. After centrifuging at 300 g 4 °C for 10 min, supernatant was collected as membrane fraction and nuclear pellet was resuspended in 0.4 ml of buffer C (1% NP-40, 500 mM Tris-HCl pH 7.4, 0.05% SDS, 20 mM EDTA, 10 mM NaF, 2 mM benzamidine) with protease inhibitors. After incubating 10 min on ice and centrifuging at 1,000 rpm 4 °C 10 min, supernatant was collected as nuclear fraction. Equal volumes of cytoplasm, membrane, and nuclear fractions were submitted to SDS-PAGE and analyzed by immunoblotting using anti-beta-tubulin (Sigma T4026, 1:1,000), HRP-conjugated anti-HA (clone 3F10 Roche 12013819001, 1:1,000), calnexin (Bethyl A303-696A, 1:2,000), lamin B1 (Abcam ab16048, 1:1,000), HRP-conjugated anti-rabbit (Santa Cruz sc-2004, 1:10,000), and HRP-conjugated anti-mouse (GE Healthcare

NA931V, 1:10,000) antibodies. Equal volumes of the three fractions were mixed and loaded as total cell extract. Uncropped blots are shown in the Source Data File.

## Isolation of detergent-resistant membranes (DRMs)

Approximately $1 \times 10^7$ cells were resuspended in 0.5 ml cold TNE buffer (25 mMTris-HCl, pH 7.5, 150 mM NaCl, 5 mM EDTA, and 1% Triton X-100; Surfact-Amps, ThermoFisher) with a tablet of protease inhibitors (Roche). Membranes were solubilized in a rotating wheel at 4 °C for 30 min. DRMs were isolated using an Optiprep™ gradient[72]: the cell lysate was adjusted to 40% Optiprep™, loaded at the bottom of a TLS.55 Beckman tube, overlaid with 600 μl of 30% Optiprep™ and 600 μl of TNE, and centrifuged for 1.5 h at 259,000 g at 4 °C. Six fractions of 400 μl were collected from top to bottom. DRMs were found in fraction 2. Equal volumes from each fraction were analyzed by SDS-PAGE and western blot analysis using HRP-conjugated anti-HA, caveolin1 (Santa Cruz sc-894, 1:500) and transferrin receptor (ThermoFisher 13-6800, 1:1,000) antibodies.

## Interactome of POU5F1B

Three technical replicates of 80 million HA-tagged POU5F1B and GFP overexpressing LS1034 and HT29 cells were used in this experiment (a total of 12 samples). Prior to affinity purification, protein expression was induced with 500 ng ml-1 of doxycycline for three days. Once subconfluent, cells were harvested in PBS 1 mM EDTA. Dry pellets were lysed in HNN lysis buffer (0.5% NP40, 50 mM HEPES pH 7.5, 150 mM NaCl, 50 mM NaF, 200 mM Na3VO4, 1 mM EDTA, supplemented with 1 mM PMSF, and protease inhibitors) and fixed with 3 mM DSP for 40 min. Reactive DSP was quenched with 100 mM Tris pH 7.5. The lysates were subjected to 250 U ml⁻¹ benzonase (Merck, 71205) for 30 min at 37 °C. Lysate was then centrifuged for 15 min at 17,000 g in order to remove insoluble material. Supernatant was then incubated with 100 μl of pre-washed anti-HA agarose beads (Sigma, A2095) for 2 h on a rotating wheel at 4 °C. Immunoprecipitates were washed three times with 2 ml and twice 1 ml HNN lysis buffer, and three times with 2 ml and twice 1 ml HNN buffer (50 mM HEPES pH 7.5, 150 mM NaCl, 50 mM NaF). Proteins were then eluted with 3 × 100 μl of 0.2 M glycine pH 2.5. Samples were neutralized and denatured with 550 μl 0.5 M NH4HCO3 pH 8.8, 6 M urea, reduced with 5 mM TCEP for 20 min at 37 °C and alkylated with 10 mM iodoacetic acid for 20 min at room temperature in the dark. Urea concentration was diluted to 1.5 M with 50 mM NH₄HCO₃ solution. Samples were then digested with 1 μg trypsin (Promega, V5113) overnight at 37 °C in the dark. The next day, trypsin digestion was stopped by lowering the pH with the addition of 50 μl of formic acid (AppliChem, A3858.0500) and peptides were purified and prepared for mass spectrometry injection at the EPFL proteomics facility as previously described[73]. All samples selected for further analyzes had to display more than 10 bait POU5F1B spectral counts in the three technical replicates in order to ensure proper bait protein levels. Only proteotypic, unique spectral counts were used. The CRAPome[74] matrix reduced list of proteins was used to subtract unspecific protein-protein interactions. Significance between bait-prey interactions was computed with the R package lsmeans. For each gene, we fitted a linear additive model with poison residues and the cell line information as a covariate. The contrast function was then used to compute p-values for differences in means. P-values were corrected for multiple testing using the Benjamini–Hochberg's method[75]. The most significant interactions were defined as having an adjusted *p* value lower than 0.01 and fold change enrichment over control bigger than 5 (Supplementary Data 2). A POU5F1B interactome was established with Cytoscape software, using the fold change to draw force–directed edges between proteins. Previously described protein-protein interactions amongst some detected preys were found in the BioGRID website or the literature.

## ERBB2 immunoprecipitation

Ten million SW480, HT29, and LS1034 GFP- and POU5F1B-expressing cells were harvested, washed with PBS, resuspended in lysis buffer (400 mM NaCl, 10 mM HEPES, pH 7.5, 0.1% NP-40, protease inhibitor cocktail –Roche–), and incubated 30 min at 4 °C with gentle agitation. The NaCl concentration was then brought down to 150 mM, samples were sonicated using a probe sonicator (three times 10 seconds at 30% amplitude) and then centrifuged at 17,000 rcf to remove insoluble material. The protein concentration of the lysates was measured using a BCA assay (Thermo Fisher, 23225), and equivalent protein quantities were incubated overnight on a rotating wheel at 4 °C with 50 μl of streptavidin agarose beads (Sigma S1638) previously bound to 3 μg of anti-ERBB2 antibody (ThermoFisher BMS120BT) for 3 h on a rotating wheel at 4 °C. Samples were then incubated three times for 5 min with 1 ml of BC100 (100 mM KCl, 10 mM Tris pH 7.8, 0.5 mM EDTA, 10% glycerol, 0.1 mM PMSF, 0.1 mM DTT), and five times with 1 ml of BC500 (BC100 with 500 mM KCl) on a rotating wheel at 4 °C. Immunoprecipitates were eluted twice in 50 μl 0.2 M glycine pH 2. Similar elution volumes and protein quantities for the IPs and the inputs, respectively, were submitted to SDS-PAGE and analyzed by immunoblotting using anti-ERBB2 (ThermoFisher MA5-13102, 1:100), HRP-conjugated anti-HA and HRP-conjugated anti-mouse antibody.

## Stable isotope labeling by amino acids in cell culture (SILAC)

SILAC experiments were performed as described[76]. POU5F1B- and GFP-overexpressing SW480 cells were grown in parallel in heavy or medium SILAC labeling mixes for 15 days (~9 cell divisions), performing a second independent experiment with inverted labeling. SILAC RPMI 1640 culture medium was complemented with 200 mg/L light L-proline, 150 mg/L heavy L-lysine (K8), and 50 mg/L heavy L-arginine (R10) for the heavy amino-acid labeling; and 200 mg/L light L-proline, 150 mg/L medium L-lysine (K4), and 50 mg/L medium L-arginine (R6) for the medium amino-acid labeling. Medium and heavy cells were combined and lysed as a mixed population. The day 15 time point was used for LC-MS/MS analysis of the total cell proteome after verifying that it allowed ~96% SILAC amino-acid incorporation rate.

For the secretome studies, exponentially growing SILAC-labeled cells were washed twice with PBS and incubated in 10 mL serum-free SILAC medium at 37 °C for 24 h. The cell death rate was below 2% (measured by trypan blue staining). Conditioned medium (CM) was collected, centrifuged at 200 g for 10 min, and filtrated through 0.22 μm membranes adding protease inhibitors (Roche). Medium and heavy labeled CM were combined at equal volumes and stored at −80 °C. CM from SILAC cells at day 15, corresponding to maximum amino acid incorporation rate, was used for further analysis. CM was concentrated using Amicon Ultra-4 and −0.5 centrifugal filters with 3 kDa cutoff (Millipore) and subjected to LC-MS/MS analysis.

The same total cell extract and secretome SILAC protocols were applied to sh-scramble, shRNA3, and shRNA5 LS1034 cells, labeled with light, medium, and heavy medium, respectively, with a second independent experiment inverting the labeling of shRNA3 and shRNA 5 cells.

Each sample was digested by Filter Aided Sample Preparation (FASP)[77] with minor modifications. Dithiothreitol (DTT) was replaced by Tris (2-carboxyethyl)phosphine (TCEP) as a reducing agent and Iodoacetamide by Chloracetamide as alkylating agent. Combined proteolytic digestion was performed using Endoproteinase Lys-C and Trypsin. Peptides were desalted on SDB-RPS StageTips[78] and dried down by vacuum centrifugation. Samples were then fractionated into 12 fractions using an Agilent OFFGEL 3100 system. The resulting fractions were desalted on SDB-RPS StageTips and dried by vacuum centrifugation. For LC MS/MS analysis, peptides were resuspended and separated by reversed-phase chromatography on a Dionex Ultimate 3000 RSLC nanoUPLC system in-line connected with an Orbitrap Lumos Fusion Mass-Spectrometer. Database search was performed

using MaxQuant (1.6.10.43)[79] against a concatenated database consisting of the UniProt human database (Uniprot release 2019_06, 74468 sequences) and common fetal bovine serum protein[80]. Carbamidomethylation was set as a fixed modification, whereas oxidation (M), phosphorylation (S,T,Y), Gln to pyro-Glu and acetylation (Protein N-term) were considered as variable modifications. SILAC quantifications were performed by MaxQuant using the standard settings with the re-quantification mode enabled. SILAC total cell (Supplementary Data 3 and 4) and secretome candidates (Supplementary Data 5 and 6) were selected when both experimental replicates showed $p$ value < 0.05 (outlier detection test as computed by MaxQuant), high intensity (above quantile 25%), and fold changes in the same direction for both replicates.

### DAVID enrichment analysis
For each RNA-seq and SILAC data set, we obtained a list of differentially expressed candidates. Differential gene expression analysis was performed using voom[81] as it has been implemented in the limma package of Bioconductor. A gene was considered to be differentially expressed when the fold change between groups was bigger than 2 and the p-value was smaller than 0.05. A moderated t-test (as implemented in the limma package of R) was used to test significance. P-values were corrected for multiple testing using the Benjamini–Hochberg's method[75]. Proteins enriched in the SILAC experiments were selected as detailed above. RNA-seq and SILAC data were plotted in R and python. The resulting candidate lists were subjected to a functional annotation chart using the online bioinformatics resource DAVID (6.8)[29].

### Kinexus antibody microarray preparation and analysis
Kinex KAM-1325 antibody microarray kits (Kinexus) were used according to the manufacturer's protocols. Each of the two arrays was loaded with protein extracts from SW480 POU5F1B- and GFP-overexpressing cells obtained from two independent lentiviral tranductions. Experts at Kinexus conducted array scanning and data collection. Data analysis was performed in-house, for a final selection of candidates based on $p$value < 0.05 (two-way Anova test), average spot intensity>1000, and average fold change between the two arrays >1.2 (and < −1.2) (Supplementary Data 7).

### Reporting summary
Further information on research design is available in the Nature Research Reporting Summary linked to this article.

## Data availability
The RNA-sequencing data generated in this study have been deposited in the GEO database under accession code GSE182467. The mass spectrometry proteomics data have been deposited to the ProteomeXchange Consortium via the PRIDE partner repository with the dataset identifiers PXD028034; PXD028035; and PXD028036. The TCGA and GTEx publicly available data used in this study are available in the dbGaP database under the accession codes phs000178.v10.p8 and phs000424.v7.p2, respectively. The raw single-cell RNA-seq and the cancer cell lines RNA-seq publicly available data used in this study are available in the European Genome–phenome Archive under accession codes EGAD00001002727 and EGAD00001000725, respectively. RNA-seq data from eighteen CRC patients were downloaded from GEO under the accession code GSE50760. RNA-seq data from H1 and H9 cell lines and the collection of hESC clones was downloaded from GEO (GSE60945, GSE83765). Long-range chromatin interactions in three CRC cell lines were obtained from Jäger, R. et al. TCGA ATAC-seq peaks were downloaded from https://gdc.cancer.gov/about-data/publications/ATACseq-AWG. Human genome hg19. Source Data are provided with this paper. All data are available in the article file, Supplementary Information and Source Data. Source data are provided with this paper.

## Code availability
Code for RNA-seq processing and Transpochimeric Gene Transcript analysis is published in the Supplementary Code file of this article: https://genome.cshlp.org/content/suppl/2021/08/16/gr.275133.120.DC1.

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

## Acknowledgements
We thank the members of the Trono lab and Maria-Eugenia Zaballa, Jose Vicente Sanchez-Mut, and Marco Cassano for stimulating discussions. We gratefully acknowledge the Proteomics Core Facility (PCF), the Histology Core Facility (HCF), the Bioimaging and Optics Platform (BIOP), the Centre of PhenoGenomics (CPG), and the Gene Expression Core Facility (GECF) at EPFL for their support & assistance in this work. This work was supported by grants from the European Research Council (KRABnKAP, No. 268721; Transpos-X, No. 694658), the Personalized Health and Related Technologies (PHRT-508) program, the Swiss National Science Foundation (310030_152879 and 310030B_173337), the Swiss Cancer League and the Aclon Foundation to D.T., and a Marie Sklodowska-Curie Fellowship to L.S.R.

## Author contributions
L.S.R. and D.T. conceived the study, interpreted the data, and wrote the manuscript; L.S.R. designed, performed and analyzed all the experiments, for some with the expert help of S.O. and L.A.; E.P., J.D., and S.D. performed the bioinformatics analyzes, A.C. the phylogenetic study, M.C.L., C.E., and S.P. the histopathological characterization of primary CRC biopsies, and C.L.A. and J.B.B. generated the SYSCOL cohort data.

## Competing interests
L.S.R., E.P., J.D., and D.T. are inventors on an international patent application (title: Transpochimeric gene trancripts (tcgts) as cancer biomarkers; identification number: US2022145395 (A1)) submitted by the École Polytechnique Fédérale de Lausanne that covers methods for transposcriptome–based biomarker discovery. The remaining authors declare no competing interests.
