## [Peer Review File · Nature Communications]

Transposon-activated POU5F1B promotes colorectal cancer growth and metastasisReviewers' Comments:

Reviewer #1:

Remarks to the Author:

In this manuscript, Laia Simó-Riudalbas and colleagues analyzed previously published human colorectal cancer RNA-seq data and identified LTR66-POU5F1B as oncogene-encoding transposable elements-driven transpochimeric gene transcript (TcGT). The authors provide compelling evidence that an LTR66 promoter drives POU5F1B expression in colorectal cancer compared to healthy colon and adenomas. Furthermore, they showed that POU5F1B fosters the proliferation and metastatic potential of colorectal cancer cells and performed multi-omics analyses to investigate the essential downstream targets of POU5F1B. Because POU5F1B TcGT expression is corelated with the prognosis of human colorectal cancer patients, their findings would have clinical value. Overall, most of the bioinformatics analyses are elegant and the results are nicely explained and commented in detail. However, there are some issues to be addressed before publication. For example, xenograft experiments and omics analyses were performed with only one cell line, which makes it difficult to generalize the results. The major/minor concerns are as follows:

Major points

1. For the cecum implantation experiments (Fig 2e and f), there is a discrepancy of the results among the bioluminescence intensity, tumor size, and tumor volume. When the tumor volume and the size are not significantly different, it would be difficult to conclude that POU5F1B overexpression enhances the tumor growth. We suggest the authors reconsider or discuss the interpretation of the results. Furthermore, we suggest the authors perform the xenograft experiments with HT29 POU5F1B cell lines (Extended Data Fig. 2) to evaluate whether or not the phenotype will be recapitulated.
2. POU5F1B Knockdown experiments are performed only in one cell line (LS1034, Extended Data Fig 2). We suggest the authors evaluate the phenotype of knockdown in another cell line such as HT55 (POU5F1B high cell line) to generalize the results. Furthermore, since the authors suggest that POU5F1B is an attractive target for novel cancer therapy (page 10), xenograft experiments to evaluate the POU5F1B knockdown effect on tumor growth and metastasis are strongly recommended.
3. The authors have beautifully performed multi-omic analyses to characterize POU5F1B-induced molecular changes. However, the downstream targets remain descriptive. We suggest the authors consider the following experiments.
A: Because most of the omics analyses are performed only in one cell line (POU5F1B-overexpression SW480), can the authors perform the analyses with another cell line (e.g., POU5F1B-overexpression HT29, POU5F1B-knockdown LS1034) or validate the candidate genes and proteins using other cell lines?
B: By integrating the omics analysis, can the authors suggest and evaluate the essential downstream of POU5F1B which is responsible for cancer cell growth and live metastasis? For example, the authors showed that POU5F1B recruit ERBB2. Whether ERBB2 are overexpressed in POU5F1B high cancers or not needs to be evaluated, and if so, for example, whether or not blocking ERBB2 partially rescues the phenotype of POU5F1B overexpression can be evaluated.
4. In Fig 6, the authors showed that the LTR66-POU5F1B TcGT is only predominant in CRC (Fig 6a, left), while POU5F1B transcript levels are also elevated in some prostate, uterus, breast, lung and stomach cancers (Fig 6a, right). We suggest the author investigate or discuss what drives the high POU5F1B expression in those cancers.

Minor points

1. In Fig 1e and f, only Stage II and III patients are included for the analyses. Given that POU5F1B is

highly expressed in Stage I, II, III, IV patients (Fig 1d), we suggest the authors include all Stage I, II, III, IV patients for the analyses.

2. Small typo in the abstract: "in spite to" should be "in spite of."

Reviewer #2:

Remarks to the Author:

The authors have conducted an interesting study of potential transposon-mediated gene misregulation in colorectal cancer samples. They identified POU5F1B as a promising candidate based on overexpression in cancer samples from multiple datasets and increased expression correlating with poor prognosis in patients and increased proliferation in vitro. They further identified POU5F1B's interaction network, enriched for proteins in cancer-relevant pathways, and claim that POU5F1B could be an important marker for additional cancer types as well. Overall I thought the study was well done and quite thorough. Below I provide a few comments to help improve the manuscript.

The long range chromatin interaction data needs to be further described and shown in fig 3a. Overall the analysis of this enhancer supporting long range interaction appears to be insufficient.

For figure 6a, the authors pointed out shorter versions of POU5F1-TcGTs are frequently observed in non-colorectal cancers. Although authors suggested other transposable elements, AluSx1, LTR33, L2b, and MLT1F1 as potential promoters, they also stated those shorter versions could be the result of incomplete reverse transcription. This should be addressed using CAGE or 5'RACE type of data to be sure.

Also in Figure 6, the results specifically mention colorectal, prostate, uterus, breast, lung, ovary, and stomach cancers have elevated expression of POU5F1B in either the TCGA samples or the collection of cancer cell lines. However, only colorectal and prostate samples show a clear difference in expression between cancer cell types and normal cell types. Uterus, breast, lung, ovary, and stomach cells all show expression in both cancer and normal cell types. It would be helpful to reorganize the subfigures to have matched cancer and normal data lined up vertically and perform a statistical test to show that these cancers have elevated expression of POU5F1B compared to normal cells. In addition, the authors should provide a more complete analysis of GTEX data, and find out how frequently the LTR66 promoter of POU5F1B-TcGT is being active across normal tissues, not just normal colorectal samples.

Reviewer #3:

Remarks to the Author:

In this manuscript, the authors identified and characterize a cancer-specific transposable element-driven gene transcripts (TcGTs), a hominid-restricted retrogene POU5F1B through aberrant activation of a primate-specific endogenous retroviral promoter. Correlating this observation, they demonstrated that the POU5F1B gene fosters the proliferation and metastatic potential of CRC cells. They went ahead and determined that POU5F1B, in spite of its phylogenetic relationship with the POU5F1/OCT4 transcription factor, is a membrane-enriched protein that interacts with ERBB2 and associated effectors, and induces intracellular signaling events and the release of trans-acting molecules involved in cell growth, angiogenesis and cell adhesion.

This is a rather provocative piece of work. If it is true, it starts to illuminate the functions of some "dark matters" in the genome. I am a bit troubled by the fact that the endogenous POU5F1B protein was never demonstrated to be detectable, this is a major shortcoming of the work. As this is the key data to show that POU5F1B does exist in cells and tumors. I am also dissatisfied with the analysis of the proteomics dataset. It seems that the analysis is simply an aggregate of the data, combining different kinds of proteomic experiments and data, but does not come to a conclusion. Several hypotheses were forthcoming, but did not validate/verify in the clinical data. Thus, in its current form, the manuscript is not suitable for publication.

Minor questions are in the following:

- 1, The authors showed LTR66-POU5F1B TcGT was a negative prognostic marker in CRC by SYSCOL data. Did LTR66-POU5F1B TcGT also had prognostic difference in other CRC datasets?
- 2, In the manuscript, experiments in Figure 2 showed POU5F1B enhanced CRC progression. In all experiments, only POU5F1B overexpressed SW480 cell line was used. POU5F1B knock down or knock out cell lines should be used to refine these experiments.
- 3, The authors declared that POU5F1B preferred to locate in cytoplasm and was enriched in membranes. This was really interesting, because OCT4 was a transcription factors and located in nucleus. As a paralog gene, based on the conclusion in manuscript, POU5F1B was different with OCT4 and played a different role in cancer. However, I think more evidence should be provided to reach this conclusion. Firstly, immunofluorescence and WB with different subcellular fraction used SW480 cells, which POU5F1B was overexpression. In Figure 4c, there was POU5F1B in nucleus. Whether artificial overexpression may affect the protein subcellular location? Whether POU5F1B needed to be activated before transferring into the nucleus should be discussed. Secondly, in the Human Protein Atlas (Science, 2015), immunofluorescent staining of HaCaT, RT4 and U-2 OS cell lines showed POU5F1B localized to the nucleoplasm. Did subcellular location of POU5F1B have cell specificity? Thirdly, Figure 4b showed IHC result in CRC cell lines. The authors should introduce the source and specificity of the antibody. Importantly, this IHC experiment should also detected in clinical CRC tissue samples. Fourthly, Figure 4d suggested that the TcGT product perhaps exerted some restructuring influence on membranes. Was there more evidence in proteomics data to prove this clue? I advise the authors to explore this question in proteomics data.
- 4, In this manuscript, the authors re-analyzed several published CRC RNA-seq datasets. Besides, they provided 4 proteomics datasets of cell lines. These proteomics datasets were new data which described the POU5F1B influence in CRC cell from multi-dimension. Disappointedly, the analysis of proteomics was too rough. Only up-regulated proteins were filtered. In Figure5, labels of key proteins on volcano plots would help readers understand. Down-regulated proteins should also be discussed. The relationship between these 4 datasets should be discussed. Importantly, the authors should link the different-regulated proteins and POU5F1B function from multi-dimension. The manuscript needs a more in-depth analysis of the data.
- 5, The author suggested that POU5F1B played pro-oncogenic effects by ERBB2-mediated signaling. Whether this conclusion can be verified in TCGA CRC data ?
- 6, Some writing errors need to be corrected. Capital and smaller letter of "ATAC-seq" in line 294. Figure 6a was partially occluded.

REVIEWER COMMENTS

Reviewer #1, expert in colorectal cancer genomics, metastasis and in vivo models:

In this manuscript, Laia Simó-Riudalbas and colleagues analyzed previously published human colorectal cancer RNA-seq data and identified LTR66-POU5F1B as oncogene-encoding transposable elements-driven transpochimeric gene transcript (TcGT). The authors provide compelling evidence that an LTR66 promoter drives POU5F1B expression in colorectal cancer compared to healthy colon and adenomas. Furthermore, they showed that POU5F1B fosters the proliferation and metastatic potential of colorectal cancer cells and performed multi-omics analyses to investigate the essential downstream targets of POU5F1B. Because POU5F1B TcGT expression is correlated with the prognosis of human colorectal cancer patients, their findings would have clinical value. Overall, most of the bioinformatics analyses are elegant and the results are nicely explained and commented in detail. However, there are some issues to be addressed before publication. For example, xenograft experiments and omics analyses were performed with only one cell line, which makes it difficult to generalize the results. The major/minor concerns are as follows:

Major points:

1. For the cecum implantation experiments (Fig 2e and f), there is a discrepancy of the results among the bioluminescence intensity, tumor size, and tumor volume. When the tumor volume and the size are not significantly different, it would be difficult to conclude that POU5F1B overexpression enhances the tumor growth. We suggest the authors reconsider or discuss the interpretation of the results.

To measure tumor growth in vivo, three million cells were subcutaneously injected in immunodeficient mice. Fresh pieces (2x2x2mm) were then excised from exponentially growing subcutaneous tumors and implanted in the caecal wall to test their metastatic potential in an orthotopic model. POU5F1B markedly enhanced the growth of tumors in the subcutaneous environment, but indeed did not exert further impact on the size of the transplanted tumor fragments. It may be that the fitness of well-constituted tumors is less crucially dependent on POU5F1B than their initial establishment and growth. Still, the orthotopic model fully confirmed

the results of intrasplenic injections, demonstrating that POU5F1B augments the metastatic power of CRC tumor cells. The text was modified accordingly (lines 121-126).

Furthermore, we suggest the authors perform the xenograft experiments with HT29 POU5F1B cell lines (Supplementary Fig. 2) to evaluate whether or not the phenotype will be recapitulated. As previously reported in the literature*, the metastatic power of HT29 CRC cell line is very high to start with, which suggests that it is not the best model to study the increase of metastasis formation upon POU5F1B overexpression *in vivo*.

*<https://www.ncbi.nlm.nih.gov/pmc/articles/PMC3207818/> [...] For HT29, all animals (8/8) developed both primary tumors and metastases. [...]

<https://www.ncbi.nlm.nih.gov/pmc/articles/PMC6860924/> [...] numerous liver metastases 4 weeks after splenic injection [...]

We thus prioritized another *in vivo* experiment rightly suggested by the reviewer, examining the impact of POU5F1B downregulation in CRC cells normally expressing this factor (see point 2 below). Furthermore, we consolidated our *in vitro* results with additional experiments of POU5F1B overexpression in SW620 cells (new Supplementary Fig. 2b).

2. POU5F1B Knockdown experiments are performed only in one cell line (LS1034, Supplementary Fig 2). We suggest the authors evaluate the phenotype of knockdown in another cell line such as HT55 (POU5F1B high cell line) to generalize the results. Furthermore, since the authors suggest that POU5F1B is an attractive target for novel cancer therapy (page 10), xenograft experiments to evaluate the POU5F1B knockdown effect on tumor growth and metastasis are strongly recommended.

Knockdown experiments in HT55 cell line were unsuccessful, as we could not deplete POU5F1B as efficiently as in LS1034 cells, and with only minimal downregulation of this factor the growth of HT55 cells was unaltered. However, we performed *in vivo* experiments with POU5F1B knockdown LS1034 cells as suggested. Of note, we used an inducible system because we could not obtain any viable POU5F1B knockout clone by CRISPR-mediated excision, and because populations constitutively knocked down for POU5F1B emerged only after a growth crisis and lots of cell death, hence did not seem to us optimal for *in vivo* experiments.

We first validated our dox-inducible knockdown system in LS1034 cells. We obtained about 80% downregulation with two different shRNAs (Supplementary Fig. 2d), which resulted in significant growth delay in cell culture (Fig. 2h). We then proceeded to evaluate tumor growth (by subcutaneous injection) and metastatic power (by intrasplenic injection) *in vivo*, using mice that were on doxycycline from a week before the injection. We observed that subcutaneous tumor growth was markedly slowed down in knocked down compared with control animals. However, the number and size of liver metastases after intrasplenic injection were not different (Supplementary Fig. 2e). This may be due to the persistence of some POU5F1B in knockdown LS1034 cells, or to the activation of a POU5F1B-independent metastasis-promoting pathway in this cell line. Of note, this correlates our finding that the LTR66-POU5F1B TcGT transcripts can be detected in many but not all CRC metastases (see Fig. 1b "cohort 2" and text lines 75-77). These additional experiments are described on lines 133-149 and illustrated in new Fig. 2h,i and Supplementary Fig. 2e.

3. The authors have beautifully performed multi-omic analyses to characterize POU5F1B-induced molecular changes. However, the downstream targets remain descriptive. We suggest the authors consider the following experiments.

A: Because most of the omics analyses are performed only in one cell line (POU5F1B-overexpression SW480), can the authors perfume the analyses with another cell line (e.g.,

POU5F1B-overexpression HT29, POU5F1B-knockdown LS1034) or validate the candidate genes and proteins using other cell lines?

Following the reviewer's advice, we beefed up our omics analysis as follows:

- RNAseq of HT29-overexpressing and control cells.
- RNAseq of inducible POU5F1B knockdown (with 2 different shRNAs)
- SILAC of inducible sh-scramble, shRNA3, and shRNA5 LS1034 cells.

This broader range of analyses allowed a consolidation and expansion of our conclusions, as discussed in our response to Reviewer 3's general comment. These results are reported in Fig. 5 and corresponding Supplementary information).

B: By integrating the omics analysis, can the authors suggest and evaluate the essential downstream of POU5F1B which is responsible for cancer cell growth and live metastasis? For example, the authors showed that POU5F1B recruit ERBB2. Whether ERBB2 are overexpressed in POU5F1B high cancers or not needs to be evaluated, and if so, for example, whether or not blocking ERBB2 partially rescues the phenotype of POU5F1B overexpression can be evaluated. No correlation of expression of POU5F1B and ERBB2 was found in the SYSCOL cohort and in the Cancer Cell Lines dataset (all tissues and CRC alone).

Furthermore, POU5F1B-overexpressing SW480 or inducible LS1034 KD displayed no differential sensitivity to Afatinib (ERBB1, 2, and 4 inhibitor) and Lapatinib (ERBB1 and 2 inhibitors), compared with their respective controls.

Finally, we attempted to evaluate the impact of ERBB2 knockdown on the POU5F1B-induced phenotype, but these experiments turned out to be technically challenging and their results were not conclusive (notably due to rapid overgrowth of cells having recuperated ERBB2 expression in spite of its shRNA-mediated knockdown). At this stage, we accordingly cannot claim that ERBB2 is an essential partner of POU5F1B, which correlates with the detection of a number of other pro-oncogenic factors in the POU5F1B interactome. We thus toned down our previously ERBB2-emphasizing comments in the discussion (lines 344-345). The figures above, as they illustrate negative results, are for reviewers only.

4. In Fig 6, the authors showed that the LTR66-POU5F1B TcGT is only predominant in CRC (Fig 6a, left), while POU5F1B transcript levels are also elevated in some prostate, uterus, breast, lung and stomach cancers (Fig 6a, right). We suggest the author investigate or discuss what drives the high POU5F1B expression in those cancers.

We indeed observed that in cancers other than CRC, the expression of POU5F1B can be driven by some other transposable element. We now confirm this observation with PacBio and CAGE data (Supplementary Fig. 6a). For us, the finding that POU5F1B-encoding TcGTs can be driven not just from LTR66 but also from other TE integrants supports the model of a strong selective pressure for expression of this protein irrespective of the underlying mechanism. We also think that this renders the elucidation of the precise mechanism at play in every tumor type, which would truly be a daunting task, less of an immediate priority.

Minor points:

1. In Fig 1e and f, only Stage II and III patients are included for the analyses. Given that POU5F1B is highly expressed in Stage I, II, III, IV patients (Fig 1d), we suggest the authors include all Stage I, II, III, IV patients for the analyses.

In Figure 1 we show the survival curve only with Stage II and III because these are the disease stages in which POU5F1B has a predictive power for survival. We cannot make conclusions out of stage I and IV because we do not have enough patients, as now explained in the text, where we nevertheless give the survival data including patients as requested by the reviewer (lines 96-99).

2. Small typo in the abstract: "in spite to" should be "in spite of."

Thank you for spotting the error, which was corrected.

Reviewer #2, expert in transposable elements:

The authors have conducted an interesting study of potential transposon-mediated gene misregulation in colorectal cancer samples. They identified POU5F1B as a promising candidate based on overexpression in cancer samples from multiple datasets and increased expression correlating with poor prognosis in patients and increased proliferation in vitro. They further identified POU5F1B's interaction network, enriched for proteins in cancer-relevant pathways, and claim that POU5F1B could be an important marker for additional cancer types as well. Overall I thought the study was well done and quite thorough. Below I provide a few comments to help improve the manuscript.

The long range chromatin interaction data needs to be further described and shown in fig 3a. Overall the analysis of this enhancer supporting long range interaction appears to be insufficient. The long-range chromatin interaction data illustrated in Fig 3a and Supplementary Fig. 3d were extracted from Jäger et al. (<https://www.nature.com/articles/ncomms7178>), in which the authors applied capture HiC approach to 14 colorectal cancer risk loci, including rs6983267, to date the most extensively studied cancer risk locus. Capture HiC was performed in LS174T, LoVo and

Colo205 colorectal cancer cell lines and the exact HiC interaction coordinates found by Jäger et al were transformed into a bed file and displayed on IGV (Supplementary Fig. 3d). We reckon that our understanding of the mechanism underlying POU5F1B expression is still superficial, but as it may at least somewhat differ from cancer to cancer we respectfully suggest that exploring this point further should be left to future studies.

For figure 6a, the authors pointed out shorter versions of POU5F1-TcGTs are frequently observed in non-colorectal cancers. Although authors suggested other transposable elements, AluSx1, LTR33, L2b, and MLT1F1 as potential promoters, they also stated those shorter versions could be the result of incomplete reverse transcription. This should be addressed using CAGE or 5'RACE type of data to be sure.

In cancers other than CRC, the expression of POU5F1B can indeed be driven by other transposable elements, a point now further documented through Supplementary Fig. 6a, which depicts PacBio and RNAseq data from breast and gastric cancer cell lines.

Also in Figure 6, the results specifically mention colorectal, prostate, uterus, breast, lung, ovary, and stomach cancers have elevated expression of POU5F1B in either the TCGA samples or the collection of cancer cell lines. However, only colorectal and prostate samples show a clear difference in expression between cancer cell types and normal cell types. Uterus, breast, lung, ovary, and stomach cells all show expression in both cancer and normal cell types. It would be helpful to reorganize the subfigures to have matched cancer and normal data lined up vertically and perform a statistical test to show that these cancers have elevated expression of POU5F1B compared to normal cells. In addition, the authors should provide a more complete analysis of GTEx data, and find out how frequently the LTR66 promoter of POU5F1B-TcGT is being active across normal tissues, not just normal colorectal samples. Following the reviewer's suggestion, we remodeled Fig. 6 in order to illustrate more convincingly how the presence of a TcGT correlates with increased levels of POU5F1B-encoding transcripts, and how cancers accordingly express more of this sequence than normal tissues.

Reviewer #3, expert in molecular biology of cancer and proteomics (Remarks to the Author):

In this manuscript, the authors identified and characterize a cancer-specific transposable element-driven gene transcripts (TcGTs), a hominid-restricted retrogene POU5F1B through aberrant activation of a primate-specific endogenous retroviral promoter. Correlating this observation, they demonstrated that the POU5F1B gene fosters the proliferation and metastatic potential of CRC cells. They went ahead and determined that POU5F1B, in spite to its phylogenetic relationship with the POU5F1/OCT4 transcription factor, is a membrane-enriched protein that interacts with ERBB2 and associated effectors, and induces intracellular signaling events and the release of trans-acting molecules involved in cell growth, angiogenesis and cell adhesion.

This is a rather provocative piece of work. If it is true, it starts to illuminate the functions of some "dark matters" in the genome. I am a bit troubled by the fact that the endogenous POU5F1B protein was never demonstrated to be detectable, this is a major shortcoming of the work.

In response to the reviewer's comment, we now provide POU5F1B-specific immunohistochemistry data on primary colorectal tumor samples, which confirm that the protein not only is present but also displays a predominantly cytoplasmic localization. A detailed description of the antibody used in these analysis is provided in our response to 'minor question 3' below.

As this is the key data to show that POU5F1B does exist in cells and tumors. I am also dissatisfied with the analysis of the proteomics dataset. It seems that the analysis is simply an aggregate of the data, combining different kinds of proteomic experiments and data, but does not come to a conclusion. Several hypotheses were forthcoming, but did not validate/verify in the clinical data. Thus, in its current form, the manuscript is not suitable for publication.

As explained in our response to reviewer 1's point 3, we now have expanded our omics data, including through the performance of SILAC coupled to LC-MS/MS of control and POU5F1B knockdown LS1034 cell extracts and supernatants. Our cumulated results point to intracellular signaling, cytoskeletal rearrangements and the secretion of growth promoting and cell-adhesion-modifying factors amongst mechanisms accounting for the pro-oncogenic action of POU5F1B. A number of leads are provided by these analyses, the follow-up of which is in progress but will still require a considerable amount of work, including biochemical and functional studies as well as clinical analyses. We respectfully suggest that this multi-pronged endeavor is beyond the scope of a manuscript, which already contains a very important sum of data.

Minor questions are in the following:

1, The authors showed LTR66-POU5F1B TcGT was a negative prognostic marker in CRC by SYSCOL data. Did LTR66-POU5F1B TcGT also had prognostic difference in other CRC datasets? We first were intrigued by the POU5F1B TcGT upon discovering that it was more frequent in metastatic samples than primary tumors of an 18-patient cohort, before embarking in the exploration of the SYSCOL data. Our conclusion is thus based on two distinct cohorts. We did not find the TcGT to be a negative prognostic marker in the TCGA dataset, but this compilation of multicentric tumor data, albeit a fantastic resource, does not constitute good material for this type of question. Not having found any publicly available material that matches the quality of the SYSCOL cohort, we have now turned to prospective studies, which we think may be best suited to assess broadly the impact of POU5F1B on the course of CRC and other tumors expressing this factor. We also hope that the publication of our study will prompt others to ask this question in the setting of clinical cohorts they may have underway.

2, In the manuscript, experiments in Figure 2 showed POU5F1B enhanced CRC progression. In all experiments, only POU5F1B overexpressed SW480 cell line was used. POU5F1B knock down or knock out cell lines should be used to refine these experiments. This very rightful suggestion led to experiments described in our response to Reviewer 1's point 2.

3, The authors declared that POU5F1B preferred to locate in cytoplasm and was enriched in membranes. This was really interesting, because OCT4 was a transcription factors and located in nucleus. As a paralog gene, based on the conclusion in manuscript, POU5F1B was different with OCT4 and played a different role in cancer. However, I think more evidence should be provided to reach this conclusion. Firstly, immunofluorescence and WB with different subcellular fraction used SW480 cells, which POU5F1B was overexpression. In Figure 4c, there was POU5F1B in nucleus. Whether artificial overexpression may affect the protein subcellular location? Whether POU5F1B needed to be activated before transferring into the nucleus should be discussed.

Secondly, in the Human Protein Atlas (Science, 2015), immunofluorescent staining of HaCaT, RT4 and U-2 OS cell lines showed POU5F1B localized to the nucleoplasm. Did subcellular location of POU5F1B have cell specificity?

Thirdly, Figure 4b showed IHC result in CRC cell lines. The authors should introduce the source and specificity of the antibody. Importantly, this IHC experiment should also be detected in clinical CRC tissue samples.

We thank the reviewers for a series of very helpful suggestions. She/he is right when suggesting that overexpression might alter the subcellular localization of a protein. However, we note that HA-tagged OCT4 and POU5F1B were expressed to similar levels in lentivector-transduced SW480 cells, yet localized exclusively in the nucleus of these cells for the former and predominantly in their cytoplasm for the latter (Fig. 4a and Supplementary Fig. 4c). We have no evidence for POU5F1B being subjected to activation-induced nuclear transport.

Subcellular location of POU5F1B does not have cell specificity, since 293T cells transiently transfected with POU5F1B or OCT4 displayed cytoplasmic/nuclear versus exclusively nuclear localizations of the overexpressed protein, similar to SW480 cells (Supplementary Fig. 4d).

The rabbit polyclonal anti-POU5F1B antibody (Sigma HPA058267) used in the Human Protein Atlas recognizes a 51 aa sequence, only 2 residues of which (underlined) differ between POU5F1B and OCT4: VGPGSEVWGIPPCPPPYELCGGMAYCGPQVGVGLVPQGGLETSQPESEAGV. We think that it detects OCT4, a hypothesis supported by the RNAseq counts detected in RT4 and U-2 OS cells, the 2 cell lines depicted in the Human Protein Atlas: RT4, 382 norm counts for POU5F1B and 8756 norm counts for OCT4; U-2 OS, 0 norm counts for POU5F1B and 3337 norm counts for OCT4.

In contrast, the anti-POU5F1B antibody used in our work is a custom-made rabbit polyclonal antibody, which we ordered from Biotem (<https://www.biotem-antibody.com/>), using as immunogen a 16 aa peptide (GDGPWGAEPGWVDPLT), 3 residues of which (underlined) differ between POU5F1B and OCT4. Furthermore, in order to enrich the serum for POU5F1B-specific antibodies, differential purification by affinity chromatography against POU5F1B and POU5F1/OCT4 was performed.

Finally, as suggested by the reviewer, we used this POU5F1B-specific antibody to perform IHC in biopsies from normal adjacent colon and primary colon adenocarcinoma from 5 CRC patients, validating the specificity of our antibody by matching IHC-scores with the POU5F1B mRNA levels measured by quantitative PCR from the same samples (Fig. 4b and Supplementary Fig. 4e). Not only did this analysis demonstrate the presence of POU5F1B in primary tumors, but it also confirmed its predominantly cytoplasmic localization.

Fourthly, Figure 4d suggested that the TcGT product perhaps exerted some restructuring influence on membranes. Was there more evidence in proteomics data to prove this clue? I advise the authors to explore this question in proteomics data.

The proteomics data indeed reveals that POU5F1B expression is associated with an upregulation of proteins involved in caveolae or associated with rafts and ruffles, as pointed to in lines 247-253 of the main text. While not a proof, this is indirect evidence that POU5F1B likely triggers some restructuring of membranes, which perhaps explains why the transferrin receptor submembrane localization is slightly altered in the presence of the retrogene product (Fig. 4d, text lines 205208).

4, In this manuscript, the authors re-analyzed several published CRC RNA-seq datasets. Besides, they provided 4 proteomics datasets of cell lines. These proteomics datasets were new data which described the POU5F1B influence in CRC cell from multi-dimension. Disappointedly, the analysis of proteomics was too rough. Only up-regulated proteins were filtered. In Figure5, labels

of key proteins on volcano plots would help readers understand. Down-regulated proteins should also be discussed. The relationship between these 4 datasets should be discussed. Importantly, the authors should link the different-regulated proteins and POU5F1B function from multi-dimension. The manuscript needs a more in-depth analysis of the data.

We expanded as suggested our omics analyses, and proceeded to a more in-depth analysis of their results, which led to a complete remodeling of Fig. 5 in Main and Supplementary information. Proteins found to be similarly deregulated in several datasets are now pointed to, factors downregulated are illustrated side-by-side with upregulated ones (although we must reckon that we did not find anything particularly insightful to say about proteins downregulated in the presence of POU5F1B). Still, we abstained from what we felt would have been an overinterpretation of the omics data to point unequivocally to pathways or mechanisms explaining the pro-oncogenic effect of POU5F1B, as several hypotheses are raised, each of which will need to be challenged through future work.

5, The author suggested that POU5F1B played pro-oncogenic effects by ERBB2-mediated signaling. Whether this conclusion can be verified in TCGA CRC data?

No correlation of expression of POU5F1B and ERBB2 was found either in the SYSCOL cohort or in the Cancer Cell Lines dataset (all tissues and CRC alone). Although the association of POU5F1B with ERBB2 and several of its known interactors leads us to believe that this RTK plays a role in the oncogenic effect of POU5F1B, at this stage we lack functional evidence to make the point (as also explained in our response to Reviewer 1's point number 3). Accordingly, we have toned down our statement about a possible link between POU5F1B and ERBB2 signaling.

6, Some writing errors need to be corrected. Capital and smaller letter of "ATAC-seq" in line 294. Figure 6a was partially occluded.

We thank the reviewer for spotting these errors, which were corrected.

With these additions, changes and explanations, we hope that our manuscript will now be found suitable for publication in Nature Communications. As a side note, although we did not make the point, we think that POU5F1B is the first described human-specific oncogene, as it is a hominid-specific retrogene, the human allele of which presents unique amino acid differences compared to its few orthologues, which our preliminary data suggests are functionally critical. While this is a whole new story, we will be very happy if the work submitted to your consideration receives a proper echo through its publication in Nature Communications.

We thank the reviewers for their constructive comments, and you for your very helpful editorial work, and look forward to hearing back from you.

Best regards,

Didier Trono and Laia Simó-Riudalbas

Reviewers' Comments:

Reviewer #1:

Remarks to the Author:

Please see attached PDF with comments and embedded figure.

In this manuscript, Laia Simó-Riudalbas and colleagues analyzed previously published human colorectal cancer RNA-seq data and identified LTR66-POU5F1B as oncogene-encoding transposable elements-driven transpochimeric gene transcript (TcGT). In this revised version of the manuscript, although the authors addressed some of the points raised by the reviewers, there still remains significant issues before considering it for publication. Importantly, xenograft experiments were performed with only one cell line, which makes it difficult to generalize the results. The major/minor concerns are as follows:

Major points

1. We still strongly suggest the authors perform the xenograft experiments with HT29 POU5F1B cell lines (Extended Data Fig. 2) to evaluate whether or not the phenotype will be recapitulated, because it is difficult to generalize the results if the experiments are performed using only one cell line. This experiment is essential to conclude that POU5F1B promotes metastasis as the POU5F1B knockdown experiments failed to prove the difference of metastatic potential *in vivo*. Although the metastatic potential of HT29 is high according to the authors and previous papers, it would be possible to evaluate the metastatic potential after POU5F1B overexpression if the authors optimize the cell numbers for transplantation and the timing of analysis. If HT29 is inappropriate to assess the metastatic potential, we suggest the authors look for another colon cancer cell line with low expression of POU5F1B and evaluate the metastatic potential of the xenograft models after overexpression of POU5F1B.
2. Likewise, *POU5F1B* Knockdown experiments are performed only in one cell line (LS1034). The authors attempted knocking down *Pou5f1b* in HT55, but observed little effect. To generalize the results, we strongly suggest that the authors perform POU5F1B knockdown in another cell line and evaluate the phenotype.
3. Because the authors showed that ERBB2 is not an essential downstream target of POU5F1B, could the authors investigate the alternative essential downstream targets of POU5F1B that play an important role? Without the investigation of further mechanism, multi-omic analyses remain descriptive.

Minor Point:

1. The authors claim that LTR66-POU5F1B TcGT is a negative prognostic marker in CRC, yet they only cite the SYSCOL cohort as evidence, while discounting the discordant TCGA data because of their claim that the TCGA data is not “high quality.” What makes the SYSCOL cohort more believable than the TCGA cohort? I would caution the authors from claiming LTR66-POU5F1B TcGT is a negative prognostic marker in CRC with their current level of evidence. The survival curve for high and low expressors of POU5F1B from the TCGA is below:

The above is inconsistent with the claim that LTR66-POU5F1B TcGT is a negative prognostic marker in CRC.

Reviewer #2:

Remarks to the Author:

I thank the authors for addressing my critiques. I don't have any new questions.

Reviewer #4:

Remarks to the Author:

I believe Reviewer #3's comments regarding proteomics data have been mostly addressed by the authors. Figure 5 now shows side by side comparison of up and downregulated proteins observed by the SILAC MS analysis. I do, however, have some comments regarding the proteomics methods. Line 686 suggests the proteins were filtered from the lysate and subjected to SILAC based LC-MS/MS analysis without any mention of how the proteins were processed to obtain peptides or any reference to previously described protocol. Line 298 – phosphoproteins were detected using the antibody. The detection could have been performed in the same mass spectrometry experiment by simply enriching the phosphopeptides. Are all candidates shown in supplementary Figure 5d phosphorylated? Line 306 – “5” has been written both alphabetically and numerically, please delete one.

Point-by-point response (in blue) to the reviewers' comments (in black).

REVIEWERS' COMMENTS

Reviewer #1, expert in colorectal cancer genomics, metastasis and in vivo models:

In this manuscript, Laia Simó-Riudalbas and colleagues analyzed previously published human colorectal cancer RNA-seq data and identified LTR66-POU5F1B as oncogene- encoding transposable elements-driven transpochimeric gene transcript (TcGT). In this revised version of the manuscript, although the authors addressed some of the points raised by the reviewers, there still remains significant issues before considering it for publication. Importantly, xenograft experiments were performed with only one cell line, which makes it difficult to generalize the results. The major/minor concerns are as follows:

Major points

We still strongly suggest the authors perform the xenograft experiments with HT29 POU5F1B cell lines (Extended Data Fig. 2) to evaluate whether or not the phenotype will be recapitulated, because it is difficult to generalize the results if the experiments are performed using only one cell line. This experiment is essential to conclude that POU5F1B promotes metastasis as the POU5F1B knockdown experiments failed to prove the difference of metastatic potential in vivo. Although the metastatic potential of HT29 is high according to the authors and previous papers, it would be possible to evaluate the metastatic potential after POU5F1B overexpression if the authors optimize the cell numbers for transplantation and the timing of analysis. If HT29 is inappropriate to assess the metastatic potential, we suggest the authors look for another colon cancer cell line with low expression of POU5F1B and evaluate the metastatic potential of the xenograft models after overexpression of POU5F1B.

We now provide both types of data requested by Reviewer 1. First, we performed intrasplenic injections with SW620 cells, another colon cancer cell line that does not express POU5F1B at baseline and was previously reported as endowed with a low metastatic potential compared with HT29 cells (Lavilla-Alonso et al. PlosOne, 2011). We first confirmed that the in vitro proliferation of these cells was increased upon overexpression of POU5F1B (Supplementary Fig. 2b). We then could demonstrate that POU5F1B-overexpressing cells yielded higher densities of liver metastases and greater numbers of extra-hepatic metastases than control cells. This new experiment is described in lines 132-135 of the manuscript, in Supplementary Figure 2d, and we modified the methods section accordingly (line 500-516).

Second, we repeated the intra-splenic injection of dox-inducible control or POU5F1B knockdown LS1034 cells. This time, we treated the cells for 72 hrs before injection, whereas we had previously injected untreated cells, hoping that the doxycycline fed to the mice would suffice to induce the knockdown and reveal a phenotype, which had not been the case. With our modified protocol, we now observed that LTR66-POU5F1B-depleted LS1034 cells displayed a decreased metastatic phenotype both in and out of the liver, compared with control, POU5F1B-high cells (manuscript lines 143-145 and Supplementary Fig. 2h). These two additional pieces of evidence thus extend and consolidate our claim that POU5F1B can act as a pro-metastatic factor in colorectal cancer.

Likewise, POU5F1B Knockdown experiments are performed only in one cell line (LS1034). The authors attempted knocking down Pou5f1b in HT55, but observed little effect. To generalize the results, we strongly suggest that the authors perform POU5F1B knockdown in another cell line and evaluate the phenotype.

We repeated the knockdown experiments in HT55 and LS174T colorectal cancer cells with two modifications in the transduction protocol: we increased the multiplicity of infection to 100 and spinoculated the cells with the viral suspension at 600g at 32 degrees for 60 minutes in the presence of polybrene. This resulted in improving levels of cell transduction and POU5F1B downregulation, which translated in an impaired growth for both cell lines (manuscript lines 137-139 and Supplementary Figure 2f,g).

Because the authors showed that ERBB2 is not an essential downstream target of POU5F1B, could the authors investigate the alternative essential downstream targets of POU5F1B that play an important role? Without the investigation of further mechanism, multi-omic analyses remain descriptive.

Due to the molecular complexity of POU5F1B-driven phenotype, we respectfully suggest that the investigation of the essential downstream targets of this oncogene will require a substantial amount of experimental work, which accordingly should be left for future studies.

Minor Point

1. The authors claim that LTR66-POU5F1B TcGT is a negative prognostic marker in CRC, yet they only cite the SYSCOL cohort as evidence, while discounting the discordant TCGA data because of their claim that the TCGA data is not "high quality." What makes the SYSCOL cohort more believable than the TCGA cohort? I would caution the authors from claiming LTR66-POU5F1B TcGT is a negative prognostic marker in CRC with their current level of evidence. The survival curve for high and low expressors of POU5F1B from the TCGA is below:

The above is inconsistent with the claim that LTR66-POU5F1B TcGT is a negative prognostic marker in CRC.

We thank the reviewer for this remark. We established survival curves for the TCGA and SYSCOL data following exactly the same procedure, with Kaplan-Meier representation of overall survival in stages II-III patients according to presence or absence of POU5F1B TcGT (Supplementary Figure 1i). This differs from the method used to draw the plot shown by Reviewer 1, where patients from all stages are divided into low and high POU5F1B expressors. Still, we did not observe the same positive correlation between presence of the TcGT and poor prognosis in the TCGA dataset. Accordingly, we modified the title of the corresponding Results subchapter to specify that we found POU5F1B to be a negative prognostic factor in the SYSCOL cohort. We also indicate that the survival data of the SYSCOL cohort differs from that of the TCGA data (line 108-111) and Supplementary Figure 1i. While we cannot explain the difference, commenting in our discussion (lines 333-335) that it "could stem from differences in patient populations and therapeutic regimens", we also observe that our finding "corroborates similar observations in hepatocellular carcinoma and gastric cancer" (lines 328-329). We add (lines 329-333) that "POU5F1B-encoding transcripts have been detected in circulating but not primary tumor cells from pancreatic ductal

adenocarcinoma (PDAC), where they have been found to be associated with a more rapid clinical deterioration, suggesting that POU5F1B contributes to conferring PDAC cells with the phenotype of circulating and, ultimately metastasis-initiating cells”.

Reviewer #2, expert in transposable elements:

I thank the authors for addressing my critiques. I don't have any new questions.

Reviewer #4, expert in proteomics:

I believe Reviewer #3's comments regarding proteomics data have been mostly addressed by the authors. Figure 5 now shows side by side comparison of up and downregulated proteins observed by the SILAC MS analysis. I do, however, have some comments regarding the proteomics methods. Line 686 suggests the proteins were filtered from the lysate and subjected to SILAC based LC-MS/MS analysis without any mention of how the proteins were processed to obtain peptides or any reference to previously described protocol.

We thank the reviewer for noticing this gap in the methodological explanation of the SILAC MS analysis. Please find detailed experimental steps in the Methods section (line 681-696). That is:

'Each sample was digested by Filter Aided Sample Preparation (FASP)⁷⁶ with minor modifications. Dithiothreitol (DTT) was replaced by Tris (2-carboxyethyl)phosphine (TCEP) as reducing agent and Iodoacetamide by Chloroacetamide as alkylating agent. A combined proteolytic digestion was performed using Endoproteinase Lys-C and Trypsin. Peptides were desalted on SDB-RPS StageTips⁷⁷ and dried down by vacuum centrifugation. Samples were then fractionated into 12 fractions using an Agilent OFFGEL 3100 system. Resulting fractions were desalted on SDB-RPS StageTips and dried by vacuum centrifugation. For LC MS/MS analysis, peptides were resuspended and separated by reversed-phase chromatography on a Dionex Ultimate 3000 RSLC nanoUPLC system in-line connected with an Orbitrap Lumos Fusion Mass-Spectrometer. Database search was performed using MaxQuant 1.6.10.4378 against concatenated database consisting of the UniProt human database (Uniprot release 2019_06, 74468 sequences) and common fetal bovine serum protein⁷⁹. Carbamidomethylation was set as fixed modification, whereas oxidation (M), phosphorylation (S,T,Y), Gln to pyro-Glu and acetylation (Protein N-term) were considered as variable modifications. SILAC quantifications were performed by MaxQuant using the standard settings with the re-quantification mode enabled. SILAC total cell (Supplementary Tables 3 and 4) and secretome candidates (Supplementary Tables 5 and 6) were selected when both experimental replicates showed p value < 0.05 (outlier detection test as computed by MaxQuant), high intensity (above quantile 25%), and fold changes in the same direction for both replicates.'

Line 298 – phosphoproteins were detected using the antibody. The detection could have been performed in the same mass spectrometry experiment by simply enriching the phosphopeptides. Are all candidates shown in supplementary Figure 5d phosphorylated?

As suggested by the reviewer, Supplementary Figure 5d was making no distinction between proteins detected with pan-specific or phospho-specific antibodies. We now represent the phospho-proteins as circles and the pan-specific detections as diamonds within the same plot, and we add the names of some candidates significantly enriched in POU5F1B overexpressing cells. More information can be found in Supplementary Table 7.

Line 306 – “5” has been written both alphabetically and numerically, please delete one.

We thank the reviewer for spotting this error, which was corrected.

Reviewers' Comments:

Reviewer #1:

Remarks to the Author:

In the revised manuscript, the authors addressed our concerns. In response to our suggestion, the authors performed intrasplenic injections with SW620 cells, another colon cancer cell line that does not express POU5F1B, and demonstrated that POU5F1B-overexpressing cells yielded higher densities of liver metastases. Further, they optimized POU5F1B knockdown experiments to evaluate the role of POU5F1B in colon cancer cell growth. Combined with the revisions made on the points raised by other reviewers, the manuscript has improved significantly. Overall, we recommend accepting it for publication in Nature Communications.

Reviewer #4:

Remarks to the Author:

The authors have satisfactorily addressed my questions regarding proteomics. I have no further comments.

Point-by-point response (in blue) to the reviewers' comments (in black).

REVIEWERS' COMMENTS

Reviewer #1, expert in colorectal cancer genomics, metastasis and in vivo models:

In this manuscript, Laia Simó-Riudalbas and colleagues analyzed previously published human colorectal cancer RNA-seq data and identified LTR66-POU5F1B as oncogene- encoding transposable elements-driven transpochimeric gene transcript (TcGT). In this revised version of the manuscript, although the authors addressed some of the points raised by the reviewers, there still remains significant issues before considering it for publication. Importantly, xenograft experiments were performed with only one cell line, which makes it difficult to generalize the results. The major/minor concerns are as follows:

Major points

We still strongly suggest the authors perform the xenograft experiments with HT29 POU5F1B cell lines (Extended Data Fig. 2) to evaluate whether or not the phenotype will be recapitulated, because it is difficult to generalize the results if the experiments are performed using only one cell line. This experiment is essential to conclude that POU5F1B promotes metastasis as the POU5F1B knockdown experiments failed to prove the difference of metastatic potential in vivo. Although the metastatic potential of HT29 is high according to the authors and previous papers, it would be possible to evaluate the metastatic potential after POU5F1B overexpression if the authors optimize the cell numbers for transplantation and the timing of analysis. If HT29 is inappropriate to assess the metastatic potential, we suggest the authors look for another colon cancer cell line with low expression of POU5F1B and evaluate the metastatic potential of the xenograft models after overexpression of POU5F1B.

We now provide both types of data requested by Reviewer 1. First, we performed intrasplenic injections with SW620 cells, another colon cancer cell line that does not express POU5F1B at baseline and was previously reported as endowed with a low metastatic potential compared with HT29 cells (Lavilla-Alonso et al. PlosOne, 2011). We first confirmed that the in vitro proliferation of these cells was increased upon overexpression of POU5F1B (Supplementary Fig. 2b). We then could demonstrate that POU5F1B-overexpressing cells yielded higher densities of liver metastases and greater numbers of extra-hepatic metastases than control cells. This new experiment is described in lines 132-135 of the manuscript, in Supplementary Figure 2d, and we modified the methods section accordingly (line 500-516).

Second, we repeated the intra-splenic injection of dox-inducible control or POU5F1B knockdown LS1034 cells. This time, we treated the cells for 72 hrs before injection, whereas we had previously injected untreated cells, hoping that the doxycycline fed to the mice would suffice to induce the knockdown and reveal a phenotype, which had not been the case. With our modified protocol, we now observed that LTR66-POU5F1B-depleted LS1034 cells displayed a decreased metastatic phenotype both in and out of the liver, compared with control, POU5F1B-high cells (manuscript lines 143-145 and Supplementary Fig. 2h). These two additional pieces of evidence thus extend and consolidate our claim that POU5F1B can act as a pro-metastatic factor in colorectal cancer.

Likewise, POU5F1B Knockdown experiments are performed only in one cell line (LS1034). The authors attempted knocking down POU5F1B in HT55, but observed little effect. To generalize the results, we strongly suggest that the authors perform POU5F1B knockdown in another cell line and evaluate the phenotype.

We repeated the knockdown experiments in HT55 and LS174T colorectal cancer cells with two modifications in the transduction protocol: we increased the multiplicity of infection to 100 and spinoculated the cells with the viral suspension at 600g at 32 degrees for 60 minutes in the presence of polybrene. This resulted in improving levels of cell transduction and POU5F1B downregulation, which translated in an impaired growth for both cell lines (manuscript lines 137-139 and Supplementary Figure 2f,g).

Because the authors showed that ERBB2 is not an essential downstream target of POU5F1B, could the authors investigate the alternative essential downstream targets of POU5F1B that play an important role? Without the investigation of further mechanism, multi-omic analyses remain descriptive.

Due to the molecular complexity of POU5F1B-driven phenotype, we respectfully suggest that the investigation of the essential downstream targets of this oncogene will require a substantial amount of experimental work, which accordingly should be left for future studies.

Minor Point

1. The authors claim that LTR66-POU5F1B TcGT is a negative prognostic marker in CRC, yet they only cite the SYSCOL cohort as evidence, while discounting the discordant TCGA data because of their claim that the TCGA data is not “high quality.” What makes the SYSCOL cohort more believable than the TCGA cohort? I would caution the authors from claiming LTR66-POU5F1B TcGT is a negative prognostic marker in CRC with their current level of evidence. The survival curve for high and low expressors of POU5F1B from the TCGA is below:

The above is inconsistent with the claim that LTR66-POU5F1B TcGT is a negative prognostic marker in CRC.

We thank the reviewer for this remark. We established survival curves for the TCGA and SYSCOL data following exactly the same procedure, with Kaplan-Meier representation of overall survival in stages II-III patients according to presence or absence of POU5F1B TcGT (Supplementary Figure 1i). This differs from the method used to draw the plot shown by Reviewer 1, where patients from all stages are divided into low and high POU5F1B expressors. Still, we did not observe the same positive correlation between presence of the TcGT and poor prognosis in the TCGA dataset. Accordingly, we modified the title of the corresponding Results subchapter to specify that we found POU5F1B to be a negative prognostic factor in the SYSCOL cohort. We also indicate that the survival data of the SYSCOL cohort differs from that of the TCGA data (line 108-111) and Supplementary Figure 1i. While we cannot explain the difference, commenting in our discussion (lines 333-335) that it “could stem from differences in patient populations and therapeutic regimens”, we also observe that our finding “corroborates similar observations in hepatocellular carcinoma and gastric cancer” (lines 328-329). We add (lines 329-333) that “POU5F1B-encoding transcripts have been detected in circulating but not primary tumor cells from pancreatic ductal adenocarcinoma (PDAC), where they have been found to be associated with a more rapid clinical deterioration, suggesting that POU5F1B contributes to conferring PDAC cells with the phenotype of circulating and, ultimately metastasis-initiating cells”.

Reviewer #2, expert in transposable elements:

I thank the authors for addressing my critiques. I don't have any new questions.

Reviewer #4, expert in proteomics:

I believe Reviewer #3's comments regarding proteomics data have been mostly addressed by the authors. Figure 5 now shows side by side comparison of up and downregulated proteins observed by the SILAC MS analysis. I do, however, have some comments regarding the proteomics methods. Line 686 suggests the proteins were filtered from the lysate and subjected to SILAC based LC-MS/MS analysis without any mention of how the proteins were processed to obtain peptides or any reference to previously described protocol.

We thank the reviewer for noticing this gap in the methodological explanation of the SILAC MS analysis. Please find detailed experimental steps in the Methods section (line 681-696). That is:

'Each sample was digested by Filter Aided Sample Preparation (FASP)⁷⁶ with minor modifications. Dithiothreitol (DTT) was replaced by Tris (2-carboxyethyl)phosphine (TCEP) as reducing agent and Iodoacetamide by Chloroacetamide as alkylating agent. A combined proteolytic digestion was performed using Endoproteinase Lys-C and Trypsin. Peptides were desalted on SDB-RPS StageTips⁷⁷ and dried down by vacuum centrifugation. Samples were then fractionated into 12 fractions using an Agilent OFFGEL 3100 system. Resulting fractions were desalted on SDB-RPS StageTips and dried by vacuum centrifugation. For LC MS/MS analysis, peptides were resuspended and separated by reversed-phase chromatography on a Dionex Ultimate 3000 RSLC nanoUPLC system in-line connected with an Orbitrap Lumos Fusion Mass-Spectrometer. Database search was performed using MaxQuant 1.6.10.4378 against concatenated database consisting of the UniProt human database (Uniprot release 2019_06, 74468 sequences) and common fetal bovine serum protein⁷⁹. Carbamidomethylation was set as fixed modification, whereas oxidation (M), phosphorylation (S,T,Y), Gln to pyro-Glu and acetylation (Protein N-term) were considered as variable modifications. SILAC quantifications were performed by MaxQuant using the standard settings with the re-quantification mode enabled. SILAC total cell (Supplementary Tables 3 and 4) and secretome candidates (Supplementary Tables 5 and 6) were selected when both experimental replicates showed p value < 0.05 (outlier detection test as computed by MaxQuant), high intensity (above quantile 25%), and fold changes in the same direction for both replicates.'

Line 298 – phosphoproteins were detected using the antibody. The detection could have been performed in the same mass spectrometry experiment by simply enriching the phosphopeptides. Are all candidates shown in supplementary Figure 5d phosphorylated?

As suggested by the reviewer, Supplementary Figure 5d was making no distinction between proteins detected with pan-specific or phospho-specific antibodies. We now represent the phospho-proteins as circles and the pan-specific detections as diamonds within the same plot, and we add the names of some candidates significantly enriched in POU5F1B overexpressing cells. More information can be found in Supplementary Table 7.

Line 306 – "5" has been written both alphabetically and numerically, please delete one.

We thank the reviewer for spotting this error, which was corrected.